


# Coccolithophore fluxes in the open tropical North Atlantic: influence of the Amazon river and of Saharan dust deposition

Catarina V. Guerreiro[1], Karl-Heinz Baumann[1,2], Geert-Jan A. Brummer[3,4], Gerhard Fischer[1,2], Laura F. Korte[3], Ute Merkel[1,2], Carolina Sá[5], Henko de Stigter[3], Jan-Berend W. Stuut[2,3]

[1] University of Bremen, Geosciences Department, Klagenfurter Str., 28359 Bremen, Germany.
[2] University of Bremen, MARUM - Center for Marine and Environmental Sciences, Leobener Str. 8, 28359 Bremen, Germany.
[3] NIOZ Royal Netherlands Institute for Sea Research, Department of Ocean Systems, Den Burg 1790 AB, and Utrecht University, The Netherlands.
[4] VU University, Earth and Climate Cluster, Department of Earth Sciences, Faculty of Earth and Life Sciences, De Boelelaan 1085 1081 HV Amsterdam, The Netherlands
[5] MARE Marine and Environmental Science Centre, Faculdade de Ciências da Universidade de Lisboa, Campo Grande, 1749-016 Lisboa, Portugal.

*Correspondence to*: Catarina V. Guerreiro (catarina.guerreiro@uni-bremen.de)

**Abstract.** Coccolithophores are calcifying phytoplankton and major contributors to both the organic and inorganic oceanic carbon pumps. Their export fluxes, species composition and seasonal patterns were determined in two sediment trap moorings in the open equatorial North Atlantic (M4 at 12ºN 49ºW and M2 at 14ºN 37ºW), which collected settling particles synchronously in successive 16-day intervals from October 2012 to November 2013, at 1200 m water depth.

The two trap locations show a similar seasonal pattern in total coccolith export fluxes and a predominantly tropical
coccolithophore settling assemblage throughout the monitored year. Species fluxes were yearlong dominated by lower photic zone (LPZ) taxa (*Florisphaera profunda*, *Gladiolithus flabellatus*), but also included upper photic zone (UPZ) taxa (*Umbellosphaera* spp., *Rhabdosphaera* spp., *Umbilicosphaera* spp., *Helicosphaera* spp.). The LPZ flora was most abundant during fall 2012, whereas the UPZ flora was more important during summer. In spite of these similarities, the western part of the study area produced persistently higher fluxes, averaging $241 \times 10^7$ coccoliths m$^{-2}$ d$^{-1}$ ($117 \times 10^7$ to $423 \times 10^7$ coccoliths m$^{-2}$ d$^{-1}$)
at station M4, compared to only $66 \times 10^7$ coccoliths m$^{-2}$ d$^{-1}$ ($25 \times 10^7$ to $153 \times 10^7$ coccoliths m$^{-2}$ d$^{-1}$) at station M2. Higher fluxes at M4 were mainly produced by the LPZ species, although most UPZ species also contributed higher fluxes, reflecting enhanced productivity in the western equatorial North Atlantic. In addition, we found two marked flux peaks of the more opportunistic species *Gephyrocapsa muellerae* and *Emiliania huxleyi* indicating a fast response to nutrient-enrichment of the UPZ, probably by wind-forced mixing, whereas increased fluxes of *G. oceanica* and *E. huxleyi* in October/November 2013
coincided with the occurrence of Amazon River affected surface waters. Since the spring and fall events of 2013 were also accompanied by two dust flux peaks we propose a scenario where atmospheric dust also provided fertilizing nutrients to this area. Enhanced surface buoyancy associated to the river plume indicates that the Amazon acted not only as a nutrient source, but also as a surface density retainer for nutrients supplied from the atmosphere. Still, lower total coccolith fluxes during these events compared to the maxima recorded in November 2012 and July 2013 indicate that transient productivity by opportunistic
species was less important than "background" tropical productivity in the equatorial North Atlantic. This study illustrates how two seemingly similar sites in an open-ocean tropical setting actually differ greatly in ecological and oceanographic terms, and provides valuable insights into the processes governing the ecological dynamics and the downward export of coccolithophores in the tropical North Atlantic.

Keywords: coccolith fluxes; phytoplankton; UPZ/LPZ; Saharan dust; Amazon River; tropical North Atlantic



## 1 Introduction

The centers of subtropical gyres and tropical open-ocean regions are marked by nearly permanently stratified waters which result in nutrient-depletion at the surface and low primary production during most of the year (see Mann and Lazier, 2006). With the exception of equatorial upwelling areas, where the renewal of nutrients in the mixed layer results from upward

advection of nutrient-rich deeper water, algal blooms in these areas are highly dependent on the seasonal dynamics of the mixed layer depth, the latter changing as a geostrophic response to the wind field and to the curl of the wind stress (e.g. Longhurst et al., 1995). Basin-scale thermocline tilting, mesoscale eddies and vertical mixing due to wind forcing and winter cooling are recognized as the main mechanisms responsible for bringing nutrients to the upper photic layer and to promote algal blooms in tropical and subtropical areas (e.g. Longhurst, 1993; Dufois et al., 2016). In addition, huge amounts of Saharan-

dust blown over and into the Atlantic Ocean every year are also thought to act as major nutrient suppliers to the nutrient-depleted equatorial North Atlantic (see Goudie and Middleton, 2001 and refs. therein; Okin et al., 2011). The fertilizing potential of Saharan dust is supported by previous studies from the Amazon Basin (Mahowald et al., 2008; 2009; Bristow et al., 2010), the Gulf of Mexico and the coast of Southern Florida (Walsh et al., 2006; Lenes et al., 2012) and from the North Atlantic subtropical gyre (Pabortsava et al., 2017). Next to Saharan dust inputs, diazotrophic fixation and reduction of $N_2$

nitrogen into new (recycled) forms of nitrogen ($NO_2^-$, $NO_3^-$ and $NH_4^+$) mostly by cyanobacteria (Carpenter et al., 1999) and the seasonal discharge and eastward advection of the nutrient-enriched Amazon River Plume (e.g. DeMaster et al., 1986) are additional nutrient sources for marine phytoplankton in the tropical Atlantic.

Despite low biological productivity per unit area, tropical oceans play an important role in the global carbon cycle because of their vastness (e.g. Wang et al., 2013; Signorini and McClain, 2012; Longhurst, 1993). How these oceanographic and

atmospheric processes are linked to phytoplankton productivity on seasonal to annual time-scales in the tropics, however, remains poorly understood. As longer-term phytoplankton sampling in the vast and remote open-ocean is rather costly, most of the available studies are based on data from snapshots taken during research cruises, or based on remote sensing estimates that only cover the phytoplankton biomass at the surface of the photic layer. Time-series sediment traps collecting settling particles (organic and inorganic) from phytoplankton export productivity over longer periods of time (from weeks to years)

offer a good alternative to plankton studies for assessing the seasonal variation of marine phytoplankton and the relative proportion of individual species or groups of species in the open ocean (e.g. Milliman, 1993; Baumann et al., 2005).

Coccolithophores, being at the same time photosynthetic and calcifying, are major contributors to the organic and inorganic oceanic carbon pumps (e.g. Rost and Riebesell, 2004), and can be studied in time-series samples collected by deep-ocean sediment traps (e.g. Broerse et al., 2000; Sprengel et al., 2002; Ziveri et al., 1995; Koebrich et al., 2015), thus providing insight

into the seasonal to inter-annual dynamics of open-ocean phytoplankton. Coccolithophores are amongst the most important phytoplankton groups within open-ocean oligotrophic waters (e.g. Winter et al., 1994) but also include more opportunistic (r-selected) taxa that quickly respond to short-term changes associated with nutrient input (e.g. Guerreiro et al., 2013). Sediment trap studies using this group are based on the assumption that the settling of coccoliths in open-ocean areas is primarily related to the production occurring in the overlaying photic layer (see Deuser et al., 1981; Honjo, 1982; Steinmetz, 1994; Beaufort

and Heussner, 2001), allowing to assess the export fluxes and seasonal trends of distinct coccolithophore species during longer periods than most of the plankton studies (Baumann et al., 2005).

Whereas a significant number of coccolithophore studies using sediment traps already exists for open-ocean areas (e.g. Knappertsbusch and Brummer, 1995; Broerse et al., 2000) and for regions near continental margins and islands (Beaufort and Heussner, 2001; Romero et al., 2002; Koebrich et al., 2015; Sprengel et al., 2002) at subtropical and temperate latitudes in the

Atlantic, there is no information available on the export and seasonal patterns of coccolithophores in the equatorial Atlantic region. Previous studies by Kinkel et al. (2000) and Winter et al. (2002) focusing on the living coccolithophore communities in the tropical Atlantic have reported *Florisphaera profunda* as the dominant species in the lower photic zone (LPZ) and *Emiliania huxleyi* as the dominant species in the upper photic zone (UPZ) close to the equatorial upwelling and in the western



Caribbean Sea, whereas *Umbellosphaera* spp. were most abundant in the UPZ of more open-ocean oligotrophic areas. Further north, in the subtropics offshore Bermuda, Haidar and Thierstein (2001) studied the relationship between the seasonal coccolithophore dynamics and several environmental parameters, including light, dissolved nutrients, salinity and temperature over a period of 3 years. Results showed that living coccolithophores display pronounced seasonal and interannual variability,

with highest standing stocks from winter to spring and lowest during summer. *E. huxleyi* was the dominant species at the surface during spring, whereas the LPZ species *F. profunda*, together with *Gladiolithus flabellatus* (referred by these authors as "*Thoracosphaera flabellata*") were most common below the surface during fall. During maximum stratification conditions of summer and fall, *Umbellosphaera* spp. increases in abundance to 75 m water depth, but in much lower cell densities than *E. huxleyi* or *F. profunda*. These previous observations were partially confirmed by Poulton et al. (2017), who defined three

depth-floral groups in the subtropical gyres and equatorial waters of the Atlantic Ocean: an upper euphotic zone flora characterized by high levels of surface radiance and high cell densities of *Umbellosphaera* spp. and holococcolithophores; a lower euphotic zone flora, characterized by intermediate to low surface radiance levels and high cell densities of *E. huxleyi* and *Gephyrocapsa ericsonii*; and a sub-euphotic zone flora, characterized by high cell densities of *F. profunda* and *Gladiolithus* spp thriving below the depth where light is considered sufficient to support photosynthesis, probably by mechanisms of

mixotrophy and/or phagotrophy. Therefore, more insight can be expected from a clear distinction between UPZ and LPZ taxa. Here, we present new data on the coccolithophore export fluxes, seasonal patterns and species composition from the open equatorial North Atlantic, to investigate the environmental factors triggering phytoplankton productivity, including Saharan dust deposition and the discharge and eastward dispersion of the Amazon River water. To assess the spatiotemporal variability of these processes, we (a) compare results from two sediment trap moorings, M2 and M4, located in the central and western

parts of the equatorial North Atlantic, respectively, and (b) relate coccolithophore data with environmental time-series data obtained from satellite remote sensing for the sediment trap sampling period and with particle flux data collected from the same sediment traps and recently published by Korte et al. (2017).

## 2 Oceanographic and meteorological settings

Surface water circulation in the study area, involving the Tropical Surface Water (TSW) and the South Atlantic Central Water

(SACW), is mostly driven by the north-easterly trade winds, responsible for generating the westward-flowing North Equatorial Current (NEC) between approximately 10º N and 20º N. South of the NEC is the North Equatorial Counter Current (NECC) flowing in eastward direction, counter to the wind (Stramma and Schott, 1999) (Fig. 1). Both currents are subjected to the latitudinal migration of the Intertropical Convergence Zone (ITCZ), between approximately 5º S and 12º N, depending on the season (Basha et al., 2015). During boreal summer, when the ITCZ is at its northernmost position, the SE trade winds intensify

along the equator, even penetrating into the northern hemisphere, and intensifying the surface circulation. Between summer and late fall, the NECC flows along an equatorial band centered between 5º N and 8º N (Longhurst, 1993; Garzoli and Katz, 1983). During the same period an important part of the northward flowing North Brazilian Current (NBC) is retroflected off the South American northeastern margin towards the east into the western tropical North Atlantic, after crossing the equator (approximately at 8º N and 50º W) (Philander, 2001), feeding into the NECC (Richardson and Walsh, 1986) and often

expanding northwards through the shedding of anticyclonic eddies drifting in NW direction (Schott et al., 1998) (Fig.1). During boreal winter, the southward migration of the ITCZ causes the weakening of the trade winds along the equator and the concomitant disappearance of the NECC in the western part of the equatorial region. This results in a generally westward-flowing circulation in the equatorial North Atlantic north of the equator west of 25ºW (Stramma and Schott, 1999). At the same time, the NBC ceases to retroflect but rather continues to flow northwestwards along the continental slope off the coast

of South America (Johns et al., 1998; Lux et al., 2001; Philander, 2001). This is the period of maximum northward transport of heat across the equator (Philander, 2001), for which the NBC is the main pathway (Goni and Johns, 2001).





The western equatorial North Atlantic where station M4 was located, is also seasonally influenced by the Amazon River, the world's largest river system which is known for discharging large volumes of fresh water into the open ocean each year (Mann and Lanzier, 2006). From August to December, when the retroflection of the NBC carries the river plume eastwards in the uppermost ~50 m of the water column, large amounts of nutrients (Boyle et al., 1977; Sholkovitz et al., 1978; De Master et al.,

1986) are entrained into the NECC far into the central equatorial North Atlantic (Muller-Karger et al., 1988; Molleri et al., 2010; Ffield, 2005) (Fig. 1). From August to October, the Amazon plume was observed to reach speeds of ~90 cm/s near 45º W with a gradual slowdown to 30 cm/s while moving eastwards. Near the African continent, a part of it continued into the Guinea Current, while another part moved northwards into the NEC and then westwards (e.g. Muller-Karger et al., 1988; Molleri et al., 2010).

The upper water masses in the study area, including the mixed layer, consist mostly of the warm, salty and nutrient-depleted TSW, in the upper ~100 m (Temperature –T: ~27º C, Salinity - S: 36.7 – 37) and the cooler, less salty and relatively nutrient-enriched South Atlantic Central Water (SACW), at depths down to ~500 m (T: 6.0 – 18º C, S: 34.3 – 35.8). Further down up to 1200 m water depth the Antarctic Intermediate Water (AAIW) is characterised by a subsurface oxygen maximum and a salinity minimum (T: 2 - 6º C, S: 33.8 - 34.8) (Emery and Meincke, 1986; Reid, 1994; Stramma and Schott, 1999). Below the

AAIW is the cold North Atlantic Deep Water (NADW) flowing southward between 1200 and 4000 m (Stramma and Schott, 1999).

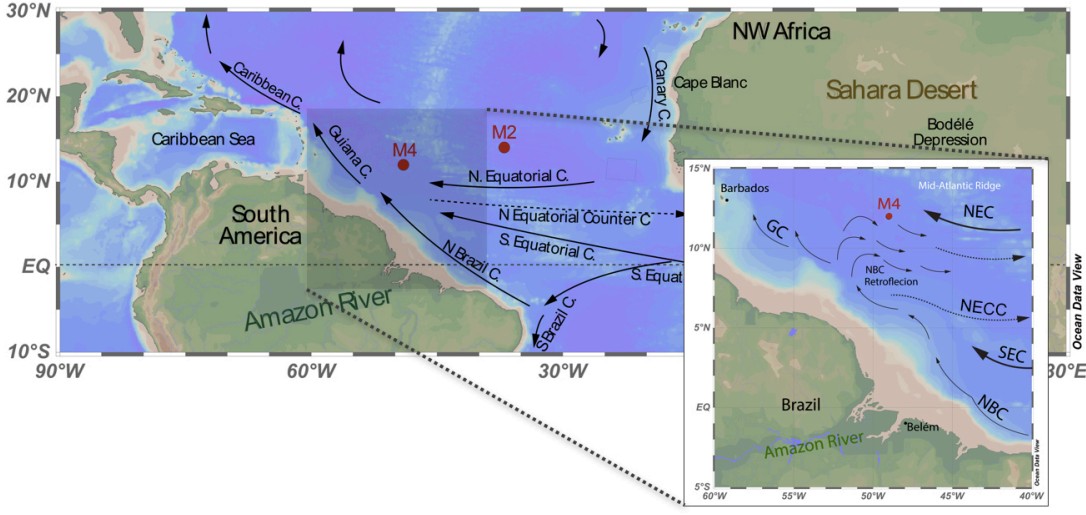

**Fig. 1. Location of the trap mooring sites M4 and M2, and schematic representation of the main surface currents in the equatorial Atlantic Ocean, with inset showing the seasonal eastward retroflection of the North Brazilian Current (NBC) during boreal summer (adapted from Mann and Lazier, 2006).**

**3 Material and methods**

**3.1. Sediment-trap sampling**

The two sediment traps used in this study, M2 (14º N, 37º W) and M4 (12º N, 49º W), collected settling particles in 24 synchronous intervals of 16 days each from 19 October 19 to 7 November 2013, at ~1200 m water depth, in the central and western equatorial North Atlantic, respectively, (Fig. 1, Table 1). Moorings were deployed during RV Meteor cruise M89 and

recovered after one year during RV Pelagia cruise 64PE378, in the context of the projects TRAFFIC and DUSTTRAFFIC (see Korte et al., 2017). Details of the mooring equipment, the deployment/recovery of the sediment traps, and the treatment of the recovered sample bottles are described in Stuut et al. (2013).



**Table 1 - Background information regarding the DUSTTRAFFIC sediment trap moorings (M4 and M2) used in this study.**

| Mooring station | M4 | M2 |
|---|---|---|
| Position | 12ºN 49º W | 14ºN 37ºW |
| Trap depth (m) | 1130 | 1200 |
| Bottom depth (m) | 5040 | 4884 |
| Sampling period | 19 Oct. 2012 – 07 Nov. 2013 | 19 Oct. 2012 – 07 Nov. 2013 |
| Number of studied samples | 23 | 23 |
| Sampling resolution (days) | 16 | 16 |

**3.2. Laboratory and microscope analysis**

Micropaleontological analysis carried out from the 1/5th split of each original sample, after centrifugal rinsing, freeze-drying and weighing (Korte et al., 2017). Each split was oxidised in a low temperature asher for approximately 4h to remove the organic matter and obtain a sample strictly composed of mineral particles (Fallet et al. 2009). The coccolith export fluxes and species composition were assessed following methods described in Andruleit (1996). Depending on the particle content of the

samples, a split of usually 1/2000 of the original sample was filtered onto polycarbonate membrane filters (47mm diameter, 0.45 μm pore size). The only exception was the material-loaded sample M4-24 for which a split of 1/16000 was necessary. Once dried, a segment of each filter was cut and mounted on a Scanning Electron Microscope (SEM) stub. A minimum of 500 coccoliths was then counted from an arbitrarily chosen transect and each coccolith was identified to the lowest taxonomic level possible at 3000 × magnification, using a Zeiss DSM 940A SEM at 10kV accelerating voltage.

Taxonomic identification of coccolithophore species followed Jordan et al. (2004) and Young et al. (2003, 2011). Coccolith species counts were converted into coccolith export fluxes (i.e. coccoliths/m$^2$/d) by extrapolating to the entire effective filter area and to the total sample, and dividing by the sample interval and the trap aperture area. Finally, the Shannon-Weaver Diversity Index H' (e.g. Tuomisto, 2013) was determined to assess the coccolithophore species diversity at each location. Shallowing/deepening of the nutricline was inferred from the ratio between upper photic zone (UPZ) species and lower photic

zone (LPZ) species, calculated as the sum of the fluxes of *Gephyrocapsa muellerae*, *G. oceanica* and *Emiliania huxleyi*, divided by the sum of the fluxes of *Florisphaera profunda* and *Gladiolithus flabellatus* (e.g. Molfino and McIntyre, 1990; Beaufort et al., 1997; Stoll et al., 2007).

**3.3. Oceanographic and meteorological data**

Time-series of hydrological (sea surface temperature – SST, salinity – SSS, and Chl-*a* concentrations) and meteorological

parameters obtained from satellite data (i.e. daily precipitation rates, Photosynthetically Available Radiance – PAR, wind speed) were used as a complementary framework for interpreting the influence of environmental variability on the coccolithophore export fluxes (Table 2). The Aerosol Optical Depth (AOD) was used as a measure of aerosols distributed within a column of air from the top of the atmosphere to the surface of the Earth. Based on Lelli et al. (2014), spatiotemporal variations of AOD during the sampling period were interpreted as an indicator for the atmospheric aerosols originating from

the African deserts. The percentage of cloud cover was estimated from MODIS data, as the percentage of pixels in the image flagged (L2-flags) with "probable cloud or ice contamination". For each trap location, satellite data were retrieved considering a 1° x 1° latitude-longitude area centred around the trap location, and averaged for each 16-day of the sediment trapping interval sediment-trap sampling period. Data were downloaded from various sources as listed in Table 2 and processed for the study period of 2012-2013, as shown in Fig. 2.






**Table 2 - List of oceanographic and meteorological parameters obtained from satellite imagery, and respective sensors, downloaded products and resolution of the downloaded data.**

| Parameter | Acronym | Units | Sensor | Product | Resolution | Source | Reference |
|---|---|---|---|---|---|---|---|
| Chlorophyll | Chl-*a* | mg m$^{-3}$ | MODIS-Aqua | L2-daily | 1 km | oceancolor.gsfc.nasa.gov | OBPG 2014; Hu et al. 2012 |
| | | | | L3-monthly | 4km | | |
| Sea Surface Temperature | SST | °C | MODIS-Aqua | L2-daily | 1 km | oceancolor.gsfc.nasa.gov | OBPG 2014; Frouin et al. 2002 & 2012 |
| Photosynthetic Available Radiation | PAR | Einstein m$^{-2}$ d$^{-1}$ | MODIS-Aqua | L2-daily | 1 km | oceancolor.gsfc.nasa.gov | OBPG 2014; Brown and Minnett (1999) |
| Aerosol Optical Depth | AOD | none | MODIS-Aqua | L2-daily | 10 km | ladsweb.nascom.nasa.gov | Levy, R., Hsu, C., et al., 2015. |
| Precipitation | Prec | mm day$^{-1}$ | multiple | L3-daily | 0.25° | disc.gsfc.nasa.gov | Goddard Earth Sciences Data and Information Services Center (2016); Huffman et al. 2007; Xie and Arkin, 1997 |
| Sea Surface Salinity and Wind Speed | SSS WSp | none m s$^{-1}$ | Aquarius | L3-daily | 1° | podaac.jpl.nasa.gov | NASA Aquarius project, 2015a & 2015b; Lee et al. 2012 |

Seasonal development of sea surface conditions did not differ drastically between the two mooring stations during the monitored period, albeit considerable differences in the range of values of salinity and Chl-*a* concentrations (Fig. 2). SST tended to be lower from late winter to early summer, and higher during late summer and fall in both areas, although generally higher at station M4 (26.2 – 29.2 ºC) than at M2 (24.3 – 28.2 ºC). SST minima were recorded in late March and April 2013 at station M4, and in late March and late May of the same year at station M2. Differences were, however, encountered in terms

of the range and seasonal patterns of SSS. Whereas station M2 revealed little variability in surface salinity (35.5 – 36.3), station M4 experienced a sharp and quite distinctive decrease during the fall months particularly in October/November 2013 (33.9 – 36.5). Surface Chl-*a* concentrations were generally low during most of the year at both stations (~ 0.1 µg l$^{-1}$) but distinctively increasing in October/November 2013 at station M4 (up to 1.13 µg l$^{-1}$) (Fig. 2ab).

Atmospheric conditions were similar at both locations. PAR values were generally higher during spring and summer (up to

65.615 Einstein m$^{-2}$ d$^{-1}$ in April 2013) and lower during fall and winter (down to 65.588 Einstein m$^{-2}$ d$^{-1}$ in December 2012) at both stations (Fig. 2cd). Precipitation rates were higher during summer and fall, up to 10 mm d$^{-1}$ in early November 2012 at M2, and up to 9 mm d$^{-1}$ in September/October 2013 at M4. Both stations received little to no precipitation during the winter and spring months (Fig. 2cd). Winds were stronger albeit variable during winter and spring, and then dropping during the summer months, to gradually increase again during the following fall. Slightly stronger winds prevailed during winter-spring

at M4 (up to 10 m/s in December 2013, April and June 2013) and during spring/summer and fall at M2 (up to 9.5 m s$^{-1}$ in May/June and October/November 2013) (Fig. 2cd). AOD had a similar month-to-month variation at both locations, generally increasing during the spring-summer transition, but reaching higher values at station M2, particularly in May/June 2013 (up to ~0.85), compared to M4 (only 0.35) (Fig. 2ef). Cloud cover percentage was fairly variable during the sampling period, generally higher at station M2 and with a tendency to increase during spring and summer. Clouds and AOD appear weakly

covariant (ascending) during winter-spring, whereas during summer-fall the discrepancy between the two parameters increases, pointing to a higher error in dust flux estimates from AOD during summer-fall (Fig. 2ef).



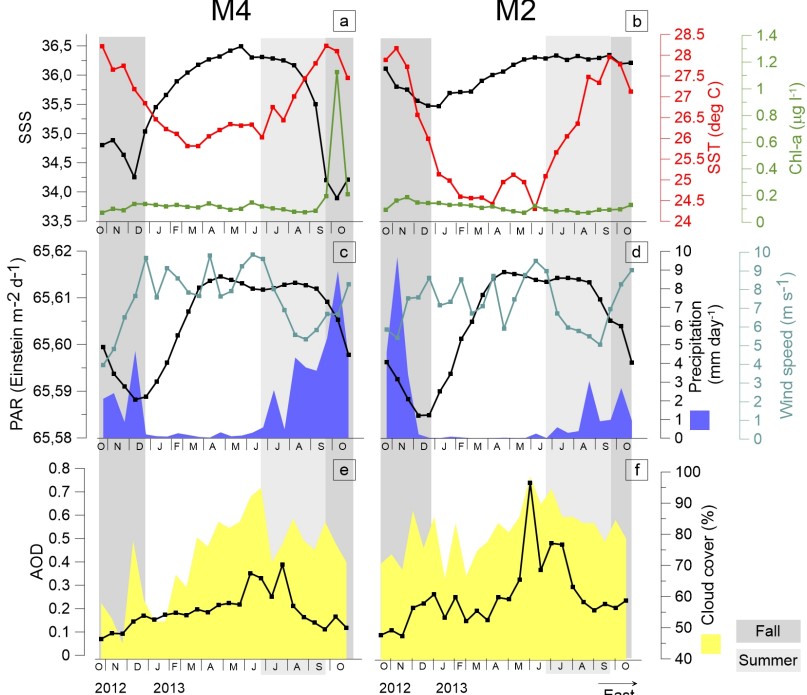

**Fig. 2. Time-series of relevant atmospheric and oceanographic parameters during the monitored time-interval, determined from remote sensing; (a-b) sea surface temperature (SST), sea surface salinity (SSS) and Chl-*a* concentrations, (c-d) photosynthetic available radiance (PAR), precipitation rate and wind speed, and (e-f) Aerosol Optical Depth (AOD) and cloud cover percentage (for data sources and processing see Tab. 2 and section 3.3). Dark and light grey bands refer to spring and summer, respectively.**

### 3.4. Statistical analysis

The relationship between the coccolithophore taxa and the environmental conditions during the monitored period was investigated on the basis of a statistical multivariate analysis (r-mode Factor Analysis, by Statistica 13) performed upon a data matrix of 46 samples (i.e. cases) and 17 variables (columns). The factor analysis considered only the percentages of the 10 commonest species/taxonomic groups (i.e. > 2% of the annual mean assemblage, in at least one of the stations), and the hydrological and meteorological parameters obtained from satellite data (SST, SSS, Chl-*a*, PAR, precipitation rate, AOD and wind speed). Results from the original data matrices were optimized through a Varimax Raw rotation and the obtained factor scores were plotted against time to assess the factors' temporal variability at both mooring locations.

### 4. Results

#### 4.1. Total coccolith fluxes at stations M4 and M2

Station M4 received much higher fluxes than M2 during most of the year, reaching a mean of $241\times10^7$ coccoliths $m^{-2}$ $d^{-1}$ ($117\times10^7$ - $423\times10^7$ coccoliths $m^{-2}$ $d^{-1}$), compared to only $66\times10^7$ coccoliths $m^{-2}$ $d^{-1}$ ($25\times10^7$ - $153\times10^7$ coccoliths $m^{-2}$ $d^{-1}$) at station M2 (Fig. 3, Table 3). In spite of this important contrast, the two stations revealed an overall similar temporal variation of coccolith fluxes, with maxima during the fall of 2012 (in early November at M4 and late October in M2) and during the summer of 2013 (in late July at M4 and early July at M2), and minima during the fall of 2012 (late October at M4 and early November at M2), spring 2013 (late March at M4 and late May at M2) and during the summer-fall transition in 2013 (at both stations). Only from January to March 2013, stations M4 and M2 diverge with fluxes below and above the corresponding annual means, respectively (Figs. 3 and 4).





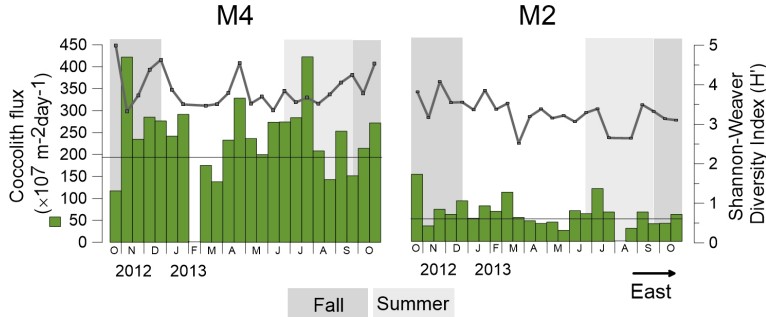

**Fig. 3 – Temporal variation of total coccolith export fluxes (coccolith m⁻² d⁻¹; green bars), Shannon-Weaver Diversity Index (H'; dashed line) and annual mean coccolith flux (horizontal black line) at trap stations M4 (left panels) and M2 (right panels).**

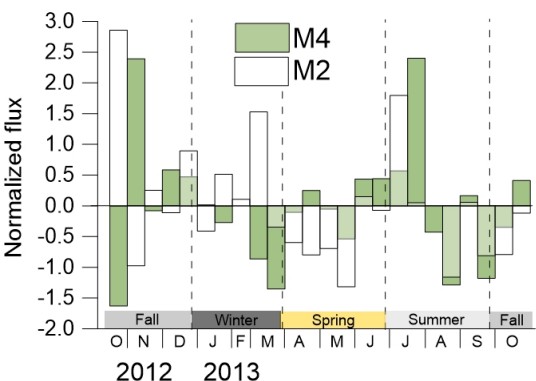

**Fig. 4 – Deviation from the annual mean coccolith flux determined for trap stations M4 (green) and M2 (transparent white).**

### 4.2. Coccolithophore species fluxes and relative abundances

The number of species/groups of species was fairly similar at stations M4 and M2 (47 and 43 taxa, respectively) although slightly higher at the westernmost site as also indicated by the higher Shannon-Weaver Diversity Index (H'; Fig. 3). H' was highest during the fall of 2012 and later in April and October/November 2013 at M4, and during late-November 2012 and lowest in March/April and July/August of 2013 at M2 (Fig. 3). In spite of this species diversity, only 10 taxa occurred in abundances ≥ 2 % in at least one of the stations and were selected for our further analysis (see Figs. 5 and 6). Mean, minimum and maximum coccolith fluxes and percentages for each taxa are given in Table 3. The dominant taxa by far were the LPZ species *Florisphaera profunda* and *Gladiolithus flabellatus*, with fluxes together comprising 74 % and 69 % of the annual mean flux at M4 and M2, respectively. The remaining taxa, including *Emiliania huxleyi*, *Umbellosphaera* spp., *Rhabdosphaera* spp., *Helicosphaera* spp., *Umbilicosphaera* spp., *Reticulofenestra sessilis*, *Gephyrocapsa muellerae* and *G. oceanica*, exhibited mean percentages of 2 - 7 % (Table 3, Fig. 5).





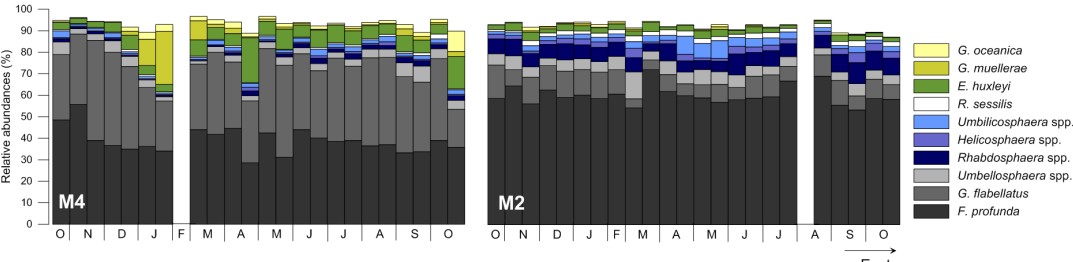

**Fig. 5 - Relative abundance of the commonest coccolithophore taxa (> 2 %) at stations M4 and M2.**

Most of the taxa produced much higher coccolith fluxes further west, at station M4. Such was particularly the case of *F.*
*profunda*, *G. flabellatus* and *E. huxleyi*, with maximum fluxes nearly 3 - 5 times higher at M4 than at M2; and the case of the
UPZ species *G. muellerae* and *G. oceanica* which recorded flux maxima up to 72 and 26 times higher at M4, respectively.
Species within *Umbilosphaera*, *Rhabdosphaera*, *Umbilicosphaera*, *Helicosphaera* and *R. sessilis* had more similar ranges of
abundance at the two sites. In terms of relative abundance, however, all the species recorded higher percentages at station M2,
compared to M4, with the exception of *G. flabellatus*, *E. huxeyi* and species within *Gephyrocapsa* spp.

**- Site M4**

Coccoliths produced by *F. profunda* (29 - 56 %) were high in November 2012 and in late July 2013, and lower fluxes in
October 2012, in late March, late August and late September 2013. In terms of relative abundances, *F. profunda* was most
common in early November 2012 and least common in late April 2013 (Fig. 6a). A similar seasonal pattern was shown by *G.*
*flabellatus* (18 – 47 %) but with a less abrupt decrease in the transition to spring compared to *F. profunda*. Lower fluxes of *G.
flabellatus* occurred in October 2012, March and October/November 2013, whereas its relative abundances generally followed
the same seasonal pattern of the fluxes (Fig. 6b). Seasonality in *Umbilosphaera* spp. (2 - 8 %) and *R. sessilis* (0.2 – 2 %) was
somewhat similar to that of *G. flabellatus*. Maximum fluxes of *Umbilosphaera* spp. occurred in December 2012 and
September 2013, lower fluxes occurred from late winter to early spring, and higher percentages in October and December
2012, and later in September 2013 (Fig. 6d). Flux maxima of *R. sessilis* occurred in late December 2012, in late July and early
October 2013, and flux minima in October 2012, late March, late April and late May 2012 (Fig. 6h).
*Rhabdosphaera* spp. (0 – 3 %) and *Umbilicosphaera* spp. (1 – 3 %) revealed similar seasonal patterns, with highest fluxes
recorded during spring and summer, and maxima in late July 2013 and later in April and June 2013 (Fig. 6e,g). In the case of
*Umbilicosphaera* spp., flux increases were also noticed in early December 2012 and November 2013. Lowest fluxes of both
taxa were recorded from January to March 2013. The seasonal pattern of their relative abundance generally followed that of
the coccolith export fluxes. Albeit some similarity with the latter taxa, *Helicosphaera* spp. (0.1 – 2 %) was clearly more
abundant in late April and late July, with lowest fluxes from late summer to late winter (Fig. 6f), whereas *E. huxleyi* (0.1 – 2
%) showed two distinctive and very sharp peaks in late April 2013 and in October/November 2013 (Fig. 6c).
*Gephyrocapsa muellerae* (0 – 25 %) occurred in persistently low fluxes and percentages throughout the year but remarkably
increased in January 2013 (Fig. 6i). Seasonality in *G. oceanica* (0.1 - 9 %) was somewhat in between *G. muellerae* and *E.
huxleyi*, but it was slightly more abundant from December 2012 until April 2013, and strikingly increasing in
October/November 2013 (Fig. 6j).

**- Site M2**

In comparison to the western site where different species revealed distinct seasonal variations, at the more central site M2 most
of the taxa revealed a very similar seasonality, with highest fluxes in late October 2012 and early July of 2013, and lower
fluxes in early November 2012, late May and August of 2013 (Fig. 6) Exceptions go to *G. flabellatus* (Fig. 6b), which was the



only one of the most abundant species without a flux increase in early March 2013, and to *Helicosphaera* spp. (Fig. 6f), which were clearly more abundant from late winter to early summer of 2013.

*F. profunda* (53 – 72 %) stood out for being much more common at site M2 compared to further west (Fig. 6a), and species within *Umbellosphaera* spp. (4 – 10 %), *H. carteri* (1 – 9 %), *Rhabdosphaera* spp. (1 – 5 %), *Umbilicosphaera* spp. (1 – 5 %),

5    *R. sessilis* (1 – 4 %) were also generally more common in this area than at M4 (Fig. 6d,f,e,g,h). By contrast, *G. flabellatus* (4 – 16 %), *E. huxleyi* (3 – 13 %), *G. oceanica* (0 – 2 %) and *G. muellerae* (0 – 1 %) revealed much lower fluxes and percentages, as well as a comparatively low month-month variation of fluxes compared to M4 (Fig. 6b,c,i,j).

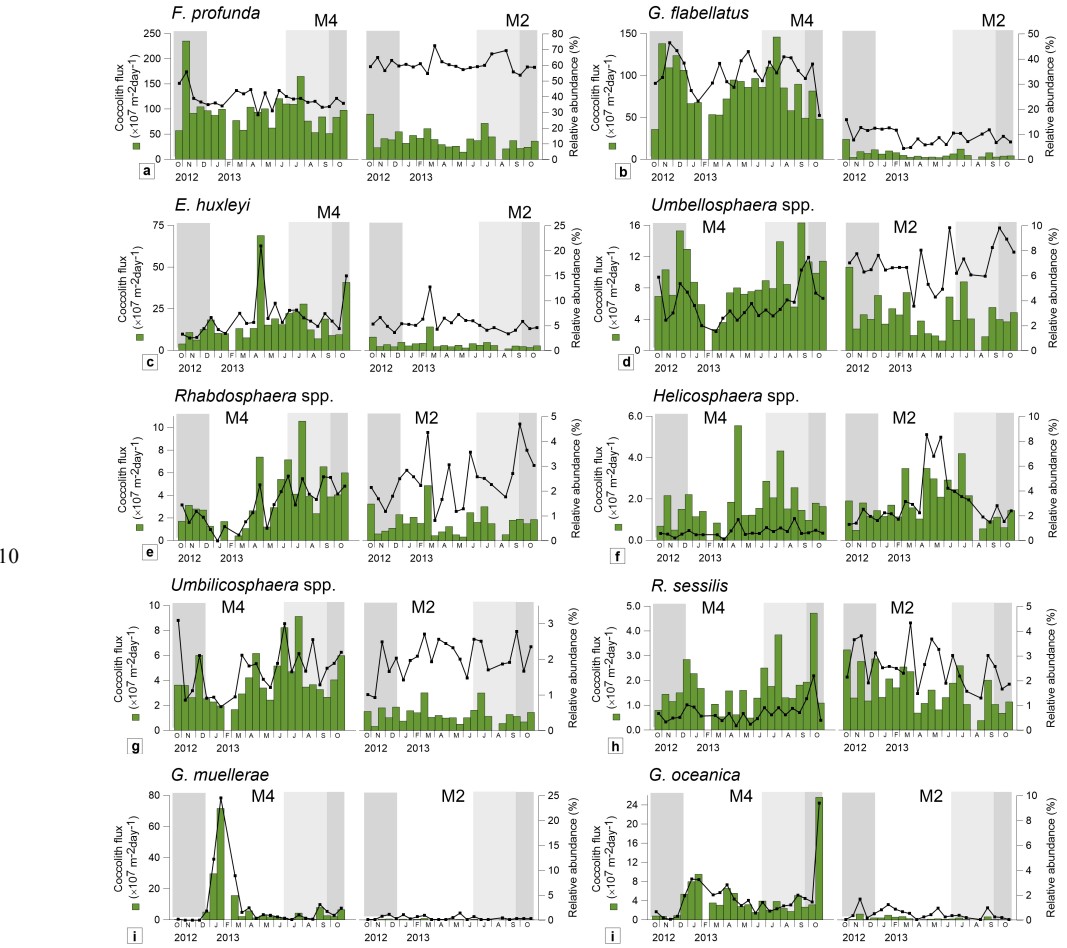

**Fig. 6 -** **Coccolith export fluxes (coccolith m⁻² d⁻¹; green bars) and relative abundance (%; black line) of the most**
15    **important species at stations M4 and M2: (a)** ***F. profunda*, (b) *G. flabellatus*, (c) *E. huxleyi*, (d) *Umbellosphaera* spp., (e)**
**Rhabdosphaera spp., (f) *Helicosphaera* spp., (g) *Umbilicosphaera* spp., (h) *R. sessilis*, (i) *G. muellerae* and (j) *G. oceanica*.**
**Dark- and light-grey bars indicate boreal fall and summer, respectively.**





**Table 3 - Annual mean and maximum coccolith fluxes and relative abundances of the most important coccolithophore species (mean >2 %), for mooring station M4 (left) and M2 (right).**

| Taxa | Coccolith fluxes (coccolith m$^{-2}$ d$^{-1}$ x 10$^7$) | | | | Relative abundances (%) | | | |
| --- | --- | --- | --- | --- | --- | --- | --- | --- |
| | **M4** | | **M2** | | **M4** | | **M2** | |
| | Mean | Min.-Max. | Mean | Min.-Max. | Mean | Min.-Max. | Mean | Min.- Max. |
| **Total coccoliths** | 247 | 117 - 423 | 66 | 25 - 153 | | | | |
| *E. huxleyi* | 17 | 4 - 69 | 4 | 1 - 14 | 7 | 3 - 21 | 5 | 3 - 13 |
| *F. profunda* | 97 | 51 - 235 | 39 | 14 - 90 | 39 | 29 - 56 | 60 | 53 - 72 |
| *G. flabellatus* | 85 | 36 - 146 | 6 | 2 - 24 | 35 | 18 - 47 | 9 | 4 - 16 |
| *G. muellerae* | 7 | 0 - 72 | 0,2 | 0 - 1 | 3 | 0 - 25 | 0,3 | 0 - 1 |
| *G. oceanica* | 4 | 0.2 - 26 | 0,3 | 0 - 1 | 2 | 0.1 - 9 | 0,4 | 0 - 2 |
| *Helicosphaera* spp. | 2 | 0.1 - 6 | 2 | 1 - 4 | 1 | 0.1 - 2 | 3 | 1 - 9 |
| *R. sessilis* | 2 | 1 - 5 | 2 | 0.4 - 3 | 1 | 0.2 - 2 | 3 | 1 - 4 |
| *Rhabdos.* spp. | 4 | 0 - 11 | 2 | 0.3 - 5 | 1 | 0 - 3 | 2 | 1 - 5 |
| *Umbellos.* spp. | 9 | 3 - 16 | 5 | 1 - 11 | 4 | 2 - 8 | 7 | 4 - 10 |
| *Umbilico.* spp. | 4 | 2 - 9 | 1 | 0.3 - 3 | 2 | 1 - 3 | 2 | 1 - 3 |

**4.4. Coccolithophore variability from multivariate factor analysis**

Four factors were extracted from the multivariate factor analysis, together explaining 63 % of the total variability within the data (Fig. 7, Table 4). Factor 1 (F1, explaining 30 % of the total variance) is represented by *F. profunda*, *R. sessilis*, *Umbellosphaera* spp., *Helicosphaera* spp. and *Rhabdosphaera* spp. in opposition to *G. flabellatus* and SST. Whereas both groups of variables do not show a clear seasonal pattern throughout the study period, they indicate contrasting ecological

conditions between M2 and M4. The first group was clearly more associated with station M2 during the entire year, particularly during spring (most negative score in late March 2013), whereas the second group was persistently more associated with station M4 (most positive scores in November 2012 and later, from May to September 2013).

Factor 2 (F2 - 16 %) is represented by precipitation, SST and Chl-*a*, in opposition to SSS, AOD and PAR, and to a lesser extent, to wind and *Helicosphaera* spp. The two groups of variables showed a similar seasonal pattern at both stations, with

precipitation, SST and Chl-*a* displaying a stronger negative signal during the fall months, particularly at M4 (most negative score in October 2013). The second group was most strongly correlated during spring and early summer at both stations (most positive scores in late-May and early-June 2013) (Fig. 7, Table 4).

Factor 3 (F3 - 10 %) is represented by *G. muellerae,* and to a lesser extent *G. oceanica*, in opposition to *Rhabdosphaera* spp., *Umbilicosphaera* spp., PAR, and to a lesser extent *Umbellosphaera* spp. The latter group revealed little statistical significance

throughout the year at either location (i.e. scores close to zero) except in January 2013, when *G. muellerae* recorded high negative scores at station M4 (Fig. 7, Table 4).

Factor 4 (F4 - 8%) reflects *F. profunda* in opposition to *E. huxleyi* and *G. oceanica,* and to a lesser extent, of wind speed. F4 reveals a weak signal at both locations during most of the year, except in April and October 2013, when two sharp positive peaks at station M4 reflected the short-term and sharp increases of *E. huxleyi* and *G. oceanica*. Relatively minor, but still

significant positive peaks occurred in January at M4 and in March at M2. Negative peaks recorded in November 2012 at both stations, and later in early May at M4, and in late August 2013 at both stations, reflecting the higher presence of *F. profunda* during these periods (Fig. 7, Table 4).



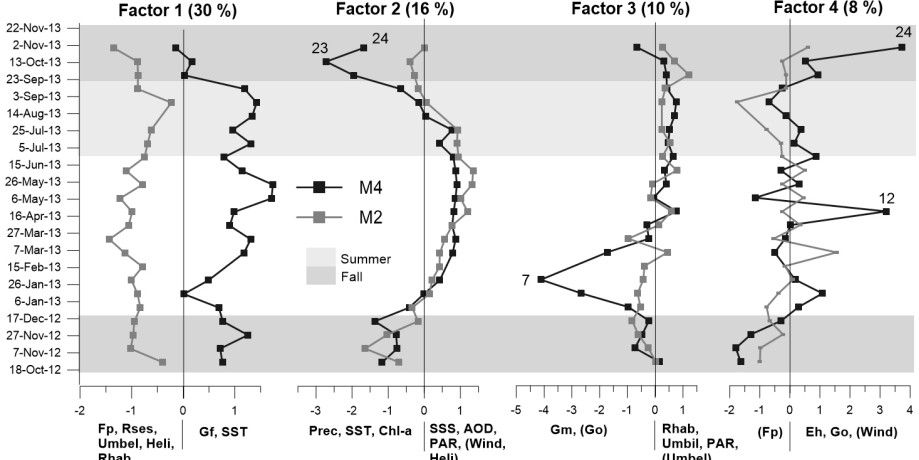

Fig. 7 – Spatiotemporal variation of the scores obtained from factor analysis. For taxonomical references, see Table 4.

Table 4 - Factor loadings (varimax raw), eigenvalues and percentage of the explained variance extracted from the data matrices referring to the period from October 2012 to October/November 2013 at stations M4 and M2 (r-mode Factor Analysis by Statistica 13; marked loadings are > 0.4).  Chl-*a* – surface phytoplankton biomass, SST and SSS - sea surface temperature and salinity, PAR – photosynthetic available radiance, AOD – aerosol optical depth, Prec – daily precipitation rate and Wind – Wind speed. In Factor 3, numbers 7 and 12 indicate the time-intervals where *G. muellerae* and *E. huxleyi* produced "pulse-like" maxima (see Fig. 6i,c), whereas in Factor 4, number 24 indicates the time-intervals where the "pulse-like" maxima of sea surface Chl-*a* and *G. oceanica*, together with a sharp increase of *E. huxleyi* occurred at station M4 (see Fig. 6c,j). Numbers 23 and 24 indicate the time-intervals when SSS minima was recorded during fall at the same site (see Fig. 2).

| | F1 | F2 | F3 | F4 |
|---|---|---|---|---|
| *F. profunda* (Fp) | -0,8 | 0,2 | 0,1 | -0,4 |
| *G. flabellatus* (Gf) | 0,9 | -0,2 | 0 | 0 |
| *E. huxleyi* (Eh) | 0,2 | 0,1 | 0,2 | 0,8 |
| *Umbellosphaera* spp. (Umbel) | -0,7 | -0,2 | 0,4 | -0,1 |
| *Rhabdosphaera* spp. (Rhab) | -0,5 | 0 | 0,6 | 0,2 |
| *Helicosphaera* spp. (Heli) | -0,6 | 0,4 | 0,2 | 0 |
| *Umbilicosphaera* spp. (Umbil) | -0,3 | 0,1 | 0,5 | 0,2 |
| *R. sessilis* (Rses) | -0,8 | 0 | 0,1 | -0,2 |
| *G. muellerae* (Gm) | 0,2 | 0,1 | -0,8 | 0,1 |
| *G. oceanica* (Go) | 0,2 | -0,2 | -0,4 | 0,6 |
| Chl-*a* | 0 | -0,5 | 0 | 0,2 |
| SST | 0,5 | -0,7 | 0,1 | -0,1 |
| PAR | 0,2 | 0,5 | 0,5 | 0,1 |
| AOD | -0,2 | 0,6 | 0,2 | 0 |
| SSS | -0,1 | 0,8 | 0,2 | -0,1 |
| Prec | 0,1 | -0,8 | 0,2 | 0 |
| Wind | 0 | 0,4 | -0,3 | 0,4 |
| Eingenvalues | 3,8 | 3,2 | 2,1 | 1,6 |
| Expl. Var. (%) | 30 | 16 | 10 | 8 |




## 5 Discussion

### 5.1. Similar temporal variations of the tropical flora in the western- and central equatorial North Atlantic

Comparable seasonal patterns in total coccolith fluxes observed at stations M4 and M2 point to similar environmental background conditions during the sampling period at both sites (Figs. 3 and 4). This is expressed in Factor 2 (explaining 16 %
of the variance), showing that the overall atmospheric and oceanographic conditions did not vary drastically between the two locations (Fig. 7). In the tropical North Atlantic meteorological conditions are mostly controlled by seasonal variations of the trade winds and of the Intertropical Convergence Zone (ITCZ), the latter being a zone of low pressure and increased cloudiness and precipitation near the equator (e.g. Oschlies and Garçon, 1998). The ITCZ migrates in latitude during summer and winter months in the Northern Hemisphere, shifting on average between 5º S during January and 12º N during July (e.g. Basha et al.,
2015) (see Fig. 8). Maxima in coccolith flux in October/November 2012 and July 2013 (Figs. 3 and 4) appear to have occurred under the direct influence of the ITCZ. This was especially the case in the fall months, as revealed by the prevailing high precipitation rates, weaker winds and low PAR conditions during this period (Fig. 2c,d and Fig. 7 - F2 negative scores). Whereas persistently high SSTs can be interpreted as an indicator of generally stratified conditions typical of tropical open-ocean regions (e.g. Mann and Lazier, 2006; Haidar and Thierstein, 2001), highest temperature values during the fall of 2012
and from mid-summer to the fall of 2013 suggest that stratification was strongest when under the influence of the ITCZ, probably reflecting the weakening of the winds during these periods. Highest coccolith fluxes recorded under these conditions suggest that the development and/or the downward settling of coccolithophores was being favoured by the higher stability of the photic layer during these periods. By contrast, minima in coccolith flux during the winter-spring period occurred when the ITCZ was displaced further south of the study area and hence, surface circulation was mostly influenced by the NE trade winds
and by the westward-flowing NEC (see section 2). This appears to be reflected in the positive correlation between wind strength and high PAR (F2 positive scores) and the negative correlation between wind and precipitation (Fig. 7). SST minima and intensification of the wind during winter-spring suggest that deepening of the mixed layer due to winter cooling combined with some wind-forced vertical mixing (Fig. 2a,b,e,f) could have resulted in some nutrient entrainment from below (see section 5.2.2). Lowest coccolith fluxes under these conditions seem to indicate that enhanced wind-forced mixing was less favourable
to the productivity and/or downward transfer of coccoliths compared to the more stable conditions of the fall and summer.

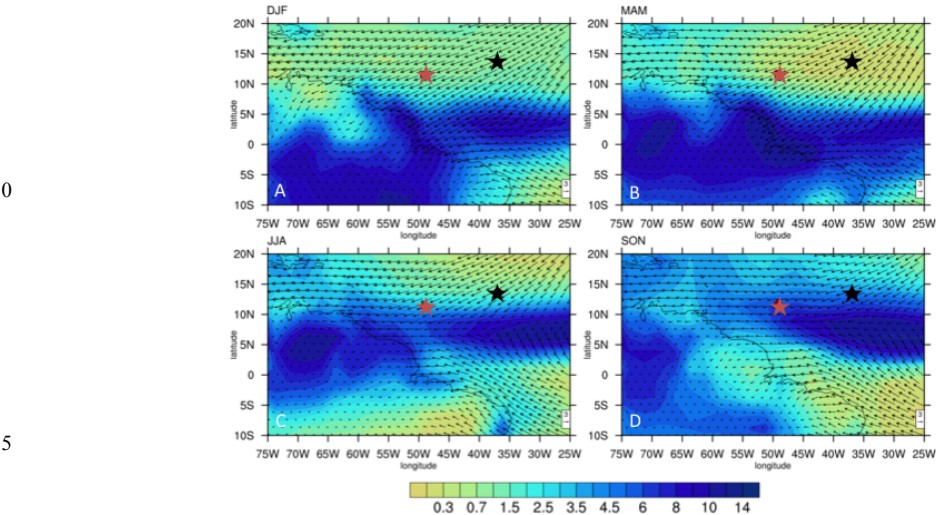

**Fig. 8. Climatological seasonal means of wind (speed and direction; reference vector length: 3 m s⁻¹) and precipitation rates (color shading; units: mm/day) over the central-western equatorial Atlantic Ocean for boreal (a) winter (Dec.-Feb.) (b) spring (March-May), (c) summer (June-Aug.), and (d) autumn (Sept.-Nov.), illustrating the seasonal**



**latitudinal migrations of the Intertropical Convergence Zone (ITCZ). Wind data (years: 1988-2015) were obtained from the *CCMP Ocean Surface Wind Vector Analyse*s (Atlas et al., 2011), and precipitation data (years: 1979-2015) from the *CPC Merged Analysis of Precipitation* (CMAP) (Xie and Arkin, 1997). Red and black stars refer to the location of sites M4 and M2, respectively.**

Comparable seasonal developments in total coccolith export fluxes at stations M2 and M4 were also reflected in the species composition. A predominantly tropical assemblage was found throughout the investigated period in both areas (Figs. 5 and 6), in general agreement with previous studies from nearby tropical and subtropical areas (Kinkel et al., 2000; Haidar and Thierstein, 2001; Winter et al., 2002; Boeckel and Baumann, 2008; Poulton et al., 2017). It included species considered well

adapted to the high-nutrient and low-light conditions prevailing in the LPZ, such as *F. profunda*, *G. flabellatus* and *R. sessilis* (e.g. Okada and Honjo, 1973; Young, 1994; Haidar and Thierstein, 2001; Winter et al., 2002), and taxa that are more often found in the nutrient-depleted and well illuminated UPZ, such as *Umbellosphaera* spp., *Rhabdosphaera* spp. and *Umbilicosphaera* spp. (e.g. Winter et al., 1994; Young, 1994; Haidar and Thierstein, 2001; Winter et al., 2002). The presence of taxa with a higher affinity for mesotrophic conditions, such as *Helicosphaera* spp. (Haidar and Thierstein, 2001; Boeckel

et al. 2006; Ziveri et al., 2004), and species with affinity with more turbulent and eutrophic environments, such as *E. huxleyi*, *G. muellerae* and *G. oceanica* (e.g. Winter et al., 1994; Kinkel et al., 2000; Winter et al., 2002; Guerreiro et al., 2013), point to occasionally enhanced environmental variability promoting nutrient input, as discussed in section 5.2.2.

In terms of seasonal patterns, however, little is known about the living coccolithophore communities thriving in the tropical Atlantic. In the subtropical Atlantic near Bermuda, 2200 km further north, seasonal variability appears more pronounced with

highest standing stocks from winter to spring and lowest during summer (Haidar and Thierstein, 2001). This is opposite to our observations in the tropical Atlantic, showing coccolith flux maxima during fall and summer (Figs. 3 and 4). Species relative proportions are different as well; whereas the subtropical Bermuda living coccolithophore community was largely dominated by the fast-blooming *E. huxleyi*, the settling coccolith assemblages at stations M4 and M2 were persistently dominated by the LPZ species *F. profunda* and *G. flabellatus* (Fig. 5). The difference between the two areas appears to reflect the much stronger

winter cooling, vertical mixing and nutrient entrainment and summer stratification in the subtropical Atlantic off Bermuda compared to the more oligotrophic and persistently stratified conditions in the tropical Atlantic (Molfino and McIntyre, 1990; Haidar and Thierstein, 2001; Mann and Lazier, 2006 and references therein). Nevertheless, higher number of coccoliths/coccospheres concentrations of *F. profunda* (~30 - 100 liths per sphere; Okada and Honjo, 1973) and possibly also of *G. flabellatus,* compared to *E. huxleyi* (~9 - 50 liths per shere; Cros and Fortuno, 2002), could result in overestimation of

its abundance in the traps. This means that the dominance of the LPZ flora in the settling coccolith assemblages from the equatorial North Atlantic may not necessarily reflect overwhelmingly higher productivity compared to *E. huxleyi*, but simply a high production rate combined with higher settling of coccoliths per cell.

That *F. profunda* and *G. flabellatus* revealed similar seasonal patterns and highest abundances during strong stratification conditions in fall and summer is consistent with observations from the open subtropical North Atlantic (Haidar and Thierstein,

2001; Broerse et al., 2000) (Figs. 6a,b). The increase of both fluxes and percentages during fall at both stations appear to reflect their better adaptation to high stratification conditions and lower light intensities compared to other species. Conversely, the increase of several UPZ tropical species within *Umbellosphaera* spp., *Rhabdosphaera* spp., *Helicosphaera* spp. and *Umbilicosphaera* spp., during summer appears to be related to slightly higher PAR levels during this period (Figs. 2c,d and 7d,g). The general affinity of *Umbellosphera* spp., *Rhabdosphaera* spp. and *Umbilicosphaera* spp. for stratified and well-

illuminated conditions was also evidenced by their higher percentages and positive correlation with PAR from late spring to early fall at both stations (F3 - Fig. 7). This is also consistent with previous studies from other tropical and subtropical areas in the North Atlantic (Haidar and Thierstein, 2001; Kinkel et al., 2000; Winter et al., 2002; Poulton et al., 2017).





### 5.2 Spatial variations in coccolith fluxes: western- vs. central equatorial North Atlantic

In spite of similar seasonal developments in both coccolith fluxes and species composition, stations M2 and M4 revealed striking differences in export fluxes by most species, pointing to the influence of environmental factors that are specific of each location. Below we highlight the main differences between the two areas and discuss the factors that potentially trigger

them.

### 5.2.1. Higher productivity and/or transfer efficiency of the LPZ flora at the western site M4

Fluxes recorded at station M4 were high, not only in comparison to station M2, but also to several other locations in the Atlantic Ocean, including open-ocean temperate and subtropical settings (e.g. Knappertsbusch and Brummer, 1995; Broerse et al., 2000; Sprengel et al., 2002), areas in the vicinity of islands (Sprengel et al., 2002), as well as more marginal regions,

even when under the influence of coastal upwelling (Beaufort and Heussner, 1999; Koebrich and Baumann, 2009). Furthermore, most of these flux studies referred to the opportunistic *E. huxleyi* as being the dominant species while the unexpectedly high total coccolith fluxes we found in the presumably oligotrophic western equatorial North Atlantic were mostly due to the LPZ dwellers *F. profunda* and *G. flabellatus*. Mean fluxes of *G. flabellatus*, in particular, were almost 14 times higher at M4 than at M2 (Figs. 6b, Table 3) and contributed importantly to Factor 1, i.e. the factor that more strikingly

distinguishes the two stations, explaining 30 % of the variance (Fig. 7). This finding is consistent with what it is known about the well constrained enhancement of this species in surface sediments from the western equatorial Atlantic and Brazilian continental margin (Kinkel et al., 2000; Boeckel et al., 2006). That *Umbellosphaera* spp., *Rhabdosphaera* spp. *Umbilicosphaera* spp., *E. huxleyi* and gephyrocapsids also produced higher coccolith fluxes in this area points to enhanced productivity throughout the entire photic zone at M4.

The persistent and overwhelming dominance of the LPZ flora suggests profiting from some sub-surface and year-round nutrient supply. Forced by the trade winds, the westward deepening of the equatorial mixed layer and associated nutricline (see Hastenrath and Merle, 1986; Longhurst, 1993; Philander, 2001) could have promoted higher production of the LPZ flora in the western equatorial Atlantic (Fig. 10). In situ CTD and nutrient measurements at both stations do indicate a consistently deeper nutricline at M4, particularly during spring, when the deep chlorophyll maximum (DCM) was found deeper at station

M4, compared to station M2 (data not shown; Roepert and Brummer in Stuut et al., 2016).

In addition to the westward tilting of the nutricline, changes in the depth range of the Antarctic Intermediate Water (AAIW) flowing in from the southwest (Reid, 1994) may have also contributed to the enhanced fluxes of the LPZ flora further west (see http://whp-atlas.ucsd.edu and Fig. 10). Originating from the surface region of the Antarctic circumpolar layer, this water mass is known to follow the South Atlantic Subtropical Gyre to enter the western equatorial Atlantic (Stramma et al., 2003;

Stramma and England, 1999). It crosses the equator and spreads along the Brazilian shelf (Talley, 1996), contributing high nutrients and low oxygen to the Gulf Stream and North Atlantic Current (Reid, 1994). Furthermore, Poulton et al. (2017) have recently reported *F. profunda* and *Gladiolithus* spp. for being able to thrive below the depth where light is thought to be sufficient to support photosynthesis in equatorial waters, probably by mechanisms of mixotrophy and/or phagotrophy. This means that these two species may be able to live even deeper in the water column than originally thought.

Higher fluxes at station M4 may also be related to higher production of faecal pellets by zooplankton grazers acting as vehicles for the downward flux of coccolithophores in this area. Recent observations by Knebel (2016) report on higher fluxes of spinose planktonic foraminifera at station M4 compared to M2. Knappertsbusch and Brummer (1995) have argued earlier that the export of coccolithophores is intimately related to day-to-day fluctuations in faecal pellet production by migrating zooplankton and nekton in the overlying mesopelagic zone. Therefore, higher zooplankton grazing in the western equatorial

North Atlantic may have increased the coccoliths' transfer efficiency hence contributing to the much higher coccolith fluxes compared to the central equatorial North Atlantic (Fig. 10).





### 5.2.2 Transient productivity of the r-selected UPZ flora at the western site M4

Spatial environmental variability in the equatorial North Atlantic is most clearly expressed by the much higher fluxes and the pulsed maxima of the more opportunistic species *E. huxleyi*, *G. muellerae* and *G. oceanica* at station M4, compared to their persistently low abundances and weak seasonality at station M2 (Figs. 6c,i,j and 5). Their occurrence in the form of short-term

high flux events in January, April and October/November 2013 at station M4 was associated with strikingly increasing ratios between these species and the LPZ flora (Fig. 9). Stoll et al. (2007) have used similar ratios as indicators of upwelling in the Bay of Bengal, based on the idea that the decrease of the LPZ flora reflects the shallowing of the nutricline (Molfino and McIntyre, 1990; Beaufort et al., 1997). The sporadic sharp increase of the UPZ/LPZ at station M4 hence suggests that the nutricline was temporarily shallower, resulting in a fast production response of the more opportunistic placolith-bearing species

to increased nutrient input. Higher fluxes of these UPZ-taxa at station M4 support the argument that the equatorial North Atlantic becomes generally more productive further west.

**- Influence of wind-forced water mixing and dispersal of the Amazon River Plume**

The pronounced maximum of *G. muellerae* in January 2013 (samples M4-6 and M4-7) (Fig. 6i and Fig. 9a) was strong enough

to be reflected in Factor 3, explaining 10 % of the variance (Fig.7). This event occurred when the wind had just started to intensify and PAR was gradually increasing, as expected from the fall-winter transition in the equatorial North Atlantic (Fig. 2). This probably resulted in some degree of vertical mixing, as suggested by the decrease of SST, which in turn may have led to the supply of nutrients from below the pycnocline to shallower levels in the photic zone (see section 5.1). The lack of a significant response by *E. huxleyi* and *G. oceanica* during this period suggests that nutrient availability was lower and/or not

persistent enough for these species to bloom. Previous studies reported *G. muellerae* to have an affinity for intermediate to higher nutrient conditions in more transitional water conditions where competition with *E. huxleyi* and *G. oceanica* is lower (e.g. Giraudeau and Bailey, 1995; Boeckel et al., 2006; Guerreiro et al., 2013), and to occupy a deeper position in the photic zone than other species of the genus Gephyrocapsa (Boeckel and Baumann, 2008; Guerreiro et al., 2013). Low surface Chl-*a* concentrations (Fig. 9a) and the absence of any significant increase of biogenic mass fluxes during this January period (Korte et al., 2017) (Fig. 9b) points to a scenario where only a few species take profit from short-term nutrient supply and moderate

light intensities at intermediate levels in the photic zone.

Similar conditions appear to have recurred in April 2013 (sample M4-12), with a pulsed maximum of *E. huxleyi* occurring under maximum PAR and lower SST, following a period of more persistent wind strength increase that lasted from early winter until late spring (Figs. 2, 6c and 9). This suggests that the water masses were stratified enough but still nutrient-enriched

due to previous wind-forced vertical mixing, hence promoting favourable conditions for blooming species to rapidly develop in the UPZ. The observed response of *E. huxleyi* is consistent with previous observations from other open-ocean areas reporting this species to dominate spring coccolithophore blooms (Haidar and Thierstein, 2001) and to induce enhanced coccolith fluxes (Broerse et al., 2000; Sprengel et al., 2002) following the water mixing and concomitant nutrient-replenishment of the photic zone, typical of late winter/early spring. That *E. huxleyi* responded in April but not in January suggests that it was more

efficiently growing under presumably higher nutrient availability and maximum light levels during spring. This is consistent with previous observations of *E. huxleyi* bursting into a bloom within only a few days, in response to nutrient availability and clear sky conditions in surface waters (Guerreiro et al., 2013). The high tolerance of this species to high levels of light has been considered crucial for its capacity to dominate coccolithophore assemblages (Nanninga and Tyrrell 1996; Tyrell and Merico, 2004). Although to a lesser extent compared to *E. huxleyi,* other taxa that also increased during this April period

include *Helicosphaera* spp., *Rhabdosphaera* spp. and *Umbilicosphaera* spp. This event was strong enough to be reflected in Factor F4, explaining 8 % of the variance (Fig. 7), in enhanced total coccolith fluxes and species H' (Fig. 3), and by a striking flux increase of $CaCO_3$, organic matter and biogenic silica (Korte et al., 2017; Fig. 9b). This suggests that nutrient and light



conditions were sufficiently favourable and persistent enough to promote the development of the entire UPZ plankton community during this short period.

The pulsed maxima of *G. oceanica* and *E. huxleyi* in October/November 2013 (sample M4-24) were also strong enough to be revealed by Factor 4 (Fig. 7) and by a slight increase of the total coccolith export production (Figs. 3 and 6c,j). Higher fluxes

of these species were accompanied by increased surface Chl-*a* concentrations (Fig.9a), pointing to high productivity in the uppermost photic zone (Figs. 6c,j and 9a). Own SEM observations of sample M4-24 revealed extremely high amounts of large diatom fragments, strongly supporting a scenario where nutrient enrichment at the surface and clear-sky conditions have promoted the development of more competitive phytoplankton species. This is furthermore supported by a striking flux increase in organic matter and biogenic silicate, and to a lesser extent of $CaCO_3$, during the same period (Korte et al., 2017;

Fig. 9b). The ability of *E. huxleyi* and *G. oceanica* to compete with diatoms in surface waters and to rapidly respond to nutrient availability was observed in coastal waters off central Portugal, confirming its capacity for rapid population growth in nutrient-rich environments (Guerreiro et al., 2013).

During the fall 2013 event, SSS had dropped to a minimum of 33.9 at station M4, compared to ~36.25 recorded at station M2 during the same period (Fig. 2a,b and 9a,b). Since both areas are under the influence of the ITCZ with comparable precipitation

regime (see Fig. 8), and given that higher precipitation during the fall of 2012 at M2 were associated with much higher SSS than at M4 for the same period (Figs. 2cd), the distinctly lower salinity at M4 in October/November 2013 was likely reflecting advected Amazon River water (Figs. 9 and 10). A strong positive correlation between high surface Chl-*a* concentrations and the salinity minimum revealed by Factor 2 (Fig. 7), and the sharp increase of surface-dwelling opportunistic species, biogenic silica and organic matter during this period (Fig. 9; Korte et al., 2017) suggest that this nutrient-rich buoyant plume was

promoting the development of phytoplankton at the surface of the western equatorial North Atlantic. The presence of the plume at station M4 was also noticed in a CTD profile taken during the recovery of the sediment trap mooring in late November 2013, showing a relatively shallow chlorophyll maximum associated with Amazon-related phytoplankton productivity (unpublished data, not shown). Previous satellite observations from this area have shown the north-eastwards dispersion of the Amazon River Plume forced by the NBC retroflection at around 5-10º N 50º W, typically from August to December, (e.g.

Muller-Karger et al., 1988; Molleri et al., 2010) (see Section 2). The plume resulted in a dramatic surface salinity contrast (Wilson et al., 1994) and promoted a gradient in environmental conditions that evolved while the plume was meandering northwards and mixing with the open-ocean waters, with a strong impact on the magnitude and composition of phytoplankton communities (refs. in Goes et al., 2014). The influence of the Amazon River water on the living coccolithophore communities has been previously reported by Winter et al. (2002) in the Caribbean Sea, where enhanced cell densities of *E. huxleyi* and *G.*

*oceanica* were seen coupled with this buoyant water mass. Our trap record clearly testifies to its impact by changing the flux and species composition of the coccolithophore communities in the oligotrophic western equatorial North Atlantic.

**- Influence of Saharan dust deposition**

Present-day deposition of Saharan dust has been recently quantified based on a transatlantic array of four sediment trap

moorings between NW Africa and the Caribbean which included mooring traps M4 and M2 (Korte et al., 2017; van der Does et al., 2016). The increase of AOD from spring to mid-summer at both stations (Fig. 2e,f), and its positive correlation with wind speed during this time (F2 positive scores, Fig. 7) corresponds to the same period when Korte et al. (2017) found the best accordance between dust outbreaks detected from satellite and high fluxes of dust-driven lithogenic particles. This agrees with previous observations that transatlantic Saharan-dust fluxes are highest during summer (e.g. Prospero et al., 2014). Since

precipitation started to increase at the beginning of July at both stations (Fig. 2cd), wet dust deposition probably contributed to the observed enhanced coccolith fluxes (Figs. 3 and 4). By exposure of the dust particles to cloud processes and mixing with anthropogenic species such as $HNO_3$ in the atmosphere, wet dust deposition is thought to provide more bioavailable (soluble) nutrients compared to dry dust deposition and hence, to have a greater fertilizing effect for primary production (Ridame et al.,





2014). With the exception of *R. sessilis*, all the tropical taxa – including both UPZ and LPZ floras – showed increased fluxes during July period at both locations, suggesting that coccolithophores had benefited from the nutrient input by dust along the entire photic zone. Positive correlation between AOD and *Helicosphaera* species thought to thrive in waters of moderately high fertility (Roth and Bergen, 1975; Haidar and Thierstein, 2001; Boeckel et al. 2006), points to an ecological response to

increased nutrient levels in July (F2 – Fig. 7). But whereas *E. huxleyi* and *G. oceanica* also slightly increased during this period, one would expect these species to have a more significant response compared to the tropical assemblage. This was however, not the case, suggesting that the dust acted more as a ballast than as a fertilizer during summer.

Two quite prominent dust flux peaks not detected by satellite were however recorded in April and in October/November of 2013 at station M4 (Korte et al., 2017), precisely when *E. huxleyi* and *G. oceanica* revealed a "pulse-like" increase in

abundance (Figs. 6c,j and 9). This points to a scenario where Saharan dust also acted as a nutrient-fertilizer in this area, in addition to surface ocean mixing (spring) and to the discharge/advection of the Amazon River Plume (fall). In fact, our observations suggest that the Amazon acted not only as a nutrient supplier during fall, but also as a buoyant surface density retainer of dust nutrients in the surface layer. Such combination would explain the much higher fluxes of biogenic silica reported for this period by Korte et al. (2017), as well as the drastic increase in Chl-*a* at the surface of the ocean (Figs. 2a and

9a). A similar case was observed offshore central Portugal, where the stabilization of a river buoyant plume was seen providing optimum stratification and nutrient-rich conditions for phytoplankton to bloom at the surface, resulting in a striking increase of Chl-*a* concentrations, *E. huxleyi*, *G. oceanica* and of long-chain diatoms within only a few days (Guerreiro et al., 2013). Given the huge amounts of Saharan dust blown continuously into and over the Atlantic Ocean (e.g. Prospero et al., 2014) one could argue that the massive algal blooms reported to occur in the western equatorial North Atlantic, until now interpreted as

being solely associated to nutrients provided by the Amazon, could actually result from the combination of surface buoyancy and nutrients provided by atmospheric dust deposition. In addition, the fall event was also marked by high precipitation rates possibly resulting in higher nutrient bioavailability by means of wet dust deposition. In contrast, during spring, not only the Amazon was not yet present in the study area, but the nearly inexistent precipitation possibly resulted in comparably lower nutrient-bioavailability by dry dust deposition (see Ridame et al., 2014). Such differences in atmospheric-sea surface

conditions between the two events apparently had an effect on phytoplankton species composition, resulting in higher development of several opportunist phytoplankton species during the fall, and only of *E. huxleyi* during spring.

The observed short-term shift from a more typically tropical (K-selected) to a more opportunistic (r-selected) settling coccolith assemblage during April and October/November 2013 at station M4 supports the dust-fertilization hypothesis earlier proposed by Martin (1990), and later corroborated by observations from the Amazon Basin (Bristow et al., 2010), Gulf of Mexico and

the coast of Southern Florida (Lenes et al., 2012). High fluxes of organic material recently observed in a sediment trap in the North Atlantic subtropical gyre (23° N 41° W) have been recently associated to enhanced phytoplankton productivity resulting from the stimulation of nitrogen fixation by Trichodesmium species following the deposition of dust-derived nutrients (Pabortsava et al., 2017). The role of aeolian dust for nitrogen-fixation in subtropical and tropical oligotrophic regions has been previously reported by several authors (e.g. Falkowsky et al., 1998; Mills et al., 2004; Jickels et al., 2005 and refs. therein;

Martino et al., 2014). In the tropical North Atlantic, where phytoplankton communities are nitrogen-limited, experiments conducted by Mills et al. (2004) have shown Saharan dust addition to stimulate nitrogen fixation in this area, presumably by supplying iron and phosphorous. This is supported by Baker et al. (2006) reporting on soluble phosphorous concentrations to be highest in Saharan aerosols from the North Atlantic. Guieu et al. (2014), on the other hand, report on the importance of strong and short-term ("pulse-like") dust deposition events on marine productivity in LNLC areas, whereas Romero et al.

(2011) add that when dust inputs are accompanied by turbulence (i.e. strong winds), its potential fertilizing effects on phytoplankton are prone to be amplified. This seem to fit quite nicely the episodic nature of the dust deposition events at station M4, and the fact that they were accompanied by some degree of wind intensification.



In spite of all these evidences, mineral dust deposition is also thought to increase carbon sequestration to the deep ocean by acting as a mineral ballast of sinking particles (Pabortsava et al., 2017). Van der Jagt et al. (submitted) reports on more abundant and faster-sinking aggregates when formed from a natural plankton community that has been exposed to Saharan dust deposition, compared to less abundant and slower-sinking aggregates when formed without dust. The same authors argue that

such dust-influenced aggregates would become heavily ballasted with lithogenic material at the surface and hence without scavenging any additional particles during their settling. This could at least partially explain why the dust peaks in spring and fall were marked by maxima of surface dwelling-species but not to maxima of the deep-dwelling species. To confirm whether the pulsed flux maxima of opportunistic species presented in our study were truly reflecting the response of living coccolithophores thriving in the overlying photic layer to such combination of factors (ecological signal), or were resulting

from enhanced particle transfer efficiency (e.g. ballasting by dust; faecal pellet production by zooplankton grazers) (Armstrong et al., 2002; Ziveri et al., 2007; Fischer and Karakas, 2009; Fischer et al., 2016) a comparison between settling coccolith assemblages and the living coccolithophore communities, as well as with in situ atmospheric-oceanographic observations would be required.

Whereas the spring and fall "r-selected pulse events" were accompanied by a slight increase in the total coccolith export, such

increase was never as high as in the peaks of November 2012 and July 2013 (see sections 4.1 and 5.1). This suggests that r-selected transient productivity is comparably less important than tropical "background" productivity for the overall bulk coccolith export production in the equatorial North Atlantic. Our observations indicate that enhanced productivity and/or transfer efficiency in the LPZ in the context of highly stratified tropical regions may be at least as important from a long-term perspective as that of fast blooming taxa more often found at higher latitudes and within productive coastal-neritic areas.

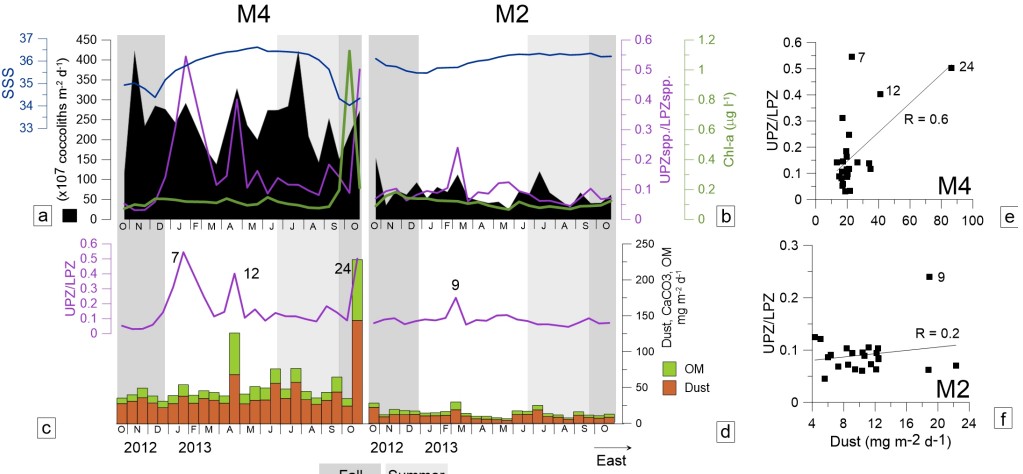

**Fig. 9.** (a-b) Seasonal variation of total coccolith fluxes (black), sea surface salinity (SSS) (dark blue), Chl-*a* concentrations (green) and UPZ/LPZ ratio (purple) calculated from the sum of the fluxes by *G. muellerae*, *G. oceanica* and *E. huxleyi*, divided by the sum of the fluxes by *F. profunda* and *G. flabellatus;* (c-d) fluxes of dust inferred from the

residual lithogenic fraction (orange bars) and organic material (OM, green bars) recently published by Korte et al. (2017); (e-f) correlation between UPZ/LPZ and the dust. Dark and light grey bands refer to spring and summer, respectively.

### 5.2.3. Typical oligotrophic open-ocean conditions at the central site M2

In contrast to station M4, persistently lower abundances of opportunistic coccolithophore species, in particular of the gephyrocapsids, and the absence of major "pulse-like" increase of these species at station M2, points to comparably more stable and oligotrophic conditions in this area. A narrower range and lower values of the UPZ/LPZ ratios at station M2 suggest that the UPZ was more persistently stratified and nutrient-depleted in the central equatorial North Atlantic (Fig. 9). An



exception occurs in early March 2013, when an increase of the UPZ/LPZ ratio at M2 and a species composition similar to that recorded in April at M4 is observed. It coincides with a slight flux increase of dust and carbonate (Korte et al., 2017), suggesting that M2 was also subjected to some degree of wind-forced water mixing (Fig. 2cd) combined with Saharan dust deposition (Fig. 9). There are no indications for influence from Amazon River water at M2, as revealed by persistently low Chl-*a*

concentrations at the surface and by low fluxes of opportunistic taxa during the fall of 2013 (Fig. 9a). Our observations suggest that, compared to M4, M2 was less affected by continental influences, be it Amazon River water or Saharan dust. The location of station M2 more central and at 2º more northerly than station M4, closer to the centre of the North Atlantic Subtropical Gyre, may have contributed to the more oligotrophic character of this station (Fig. 10-G).

That the evidence we found for ocean fertilization by Saharan dust are more significant at station M4 than at station M2 may

be due to the presence of buoyant Amazon water retaining nutrients and phytoplankton near the surface at M4, as well as to higher wet dust deposition during the fall 2013 at station M4 (e.g. Ridame et al., 2014) (Fig. 2cd). Higher distance to the dust source in Africa at M4 compared to M2 may have also contributed for increasing the bioavailability of dust-driven nutrients further west (see Stuut and Prins, 2014). This could also help explaining the lack of clear evidence for dust fertilization in open-ocean regions west of Africa (see Fischer et al., 2016; Neuer et al., 2004), since aerosols sinking in these areas would

not be fine and chemically processed enough to act as fertilizers.

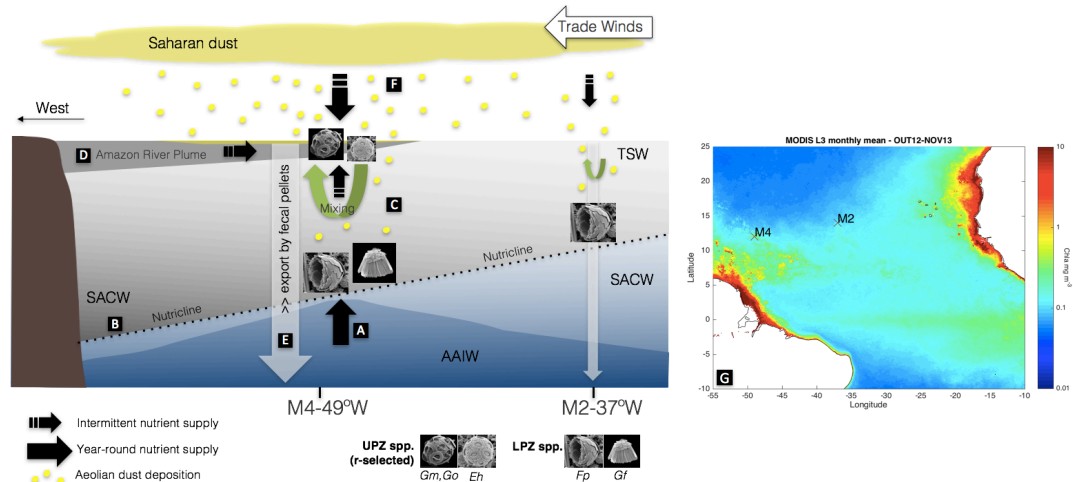

**Fig. 10.** Schematic figure summarizing the main environmental mechanisms interpreted as being at the origin of the ecological contrasts observed between stations M4 and M2 from October 2012 - October/November 2013: **(A)** nutrient-supply by AAIW depth range oscillations (dark blue lowermost layer) combined with **(B)** nutricline E-W basin-scale tilting (dashed black line), promoting the development of the LPZ species *G. flabellatus* and *F. profunda* further west;
**(C)** wind-forced surface ocean mixing and **(D)** eastward dispersion of the Amazon River Plume, resulting in transient "pulse-like" increases of the opportunistic UPZ species *E. huxleyi*, *G. muellerae* and *G. oceanica*; **(E)** higher production of fecal pellets by zooplankton grazers (white vertical arrow) at M4 contributing to increase the coccolith export efficiency; **(F)** Saharan dust deposition (yellow dots) influencing the study area, but particularly at M4 where the Amzon River Plume acts as a surface density retainer of nutrients settling from the atmosphere; **(G)** sea surface Chl-*a*
concentrations averaged for the sampling period in the equatorial Atlantic Ocean illustrating the contrast between higher and lower surface productivity at stations M4 and M2, respectively. Surface Abbreviations: UPZ – upper photic zone; LPZ – lower photic zone; *E. huxleyi*—Eh; *G. muellerae*—Gm; *G. oceanica*—Go; Fp – *F. profunda*; Gf – *G. flabellatus;* TSW – Tropical Surface Water; SACW – South Atlantic Central Water; AAIW – Antarctic Intermediate Water.




## 6. Conclusions

Our study provides important insights into the environmental factors governing the spatiotemporal variability of coccolithophores in the equatorial North Atlantic, and illustrates how this supposedly oligotrophic and stable open-ocean region actually reveals significant ecological variability. The main findings from our sediment-trap study are as follows:

1.  A predominantly tropical coccolith settling assemblage and a generally similar seasonality in total coccolith fluxes at the western station M4 and central station M2 points to comparable background environmental conditions at both sites. Flux maxima were associated with stronger stratification conditions under the influence of the Intertropical Convergence Zone (ITCZ) during summer-fall, whereas flux minima occurred during stronger NE trade winds and lower SSTs during winter-spring.

2.  Low light and deep nutricline dwelling *F. profunda* and *G. flabellatus* dominated at both locations during the entire sampling period but especially during fall. Upper photic zone taxa such as *Umbellosphaera* spp., *Rhabdosphaera* spp., *Helicosphaera* spp. and *Umbilicosphaera* spp. were more abundant during summer.

3.  In spite of the similar seasonal pattern, the two open-ocean locations in the oligotrophic equatorial North Atlantic revealed striking differences in coccolith export fluxes, species' proportions and oceanographic
    processes.

4.  Total coccolith fluxes were almost 4 times higher at the western station M4 than at the central station M2, mostly due to *G. flabellatus* and *F. profunda,* although many other species showed higher fluxes year-round. We interpret this as resulting from (i) westward deepening of the north equatorial nutricline and nutrient supply by the Antarctic Intermediate Water (AAIW), both favouring the LPZ flora, and (ii) higher transfer
    efficiency, possibly by enhanced zooplankton grazing in this area.

5.  Higher abundances and pulsed flux maxima of more opportunistic species at station M4 point to the occurrence of transient productivity in this area: (a) the increase of *E. huxleyi* in April reflected a fast ecological response to intermittent nutrient input promoted by sea surface cooling and wind-forced vertical mixing, combined with dry dust deposition, whereas (b) pulsed maxima of *E. huxleyi* and *G. oceanica* in
    October/November reflected a response to surface nutrient-enrichment derived from Amazon River discharge combined with wet dust deposition.

6.  Enhanced surface buoyancy provided by the relatively less saline Amazon River plume appears to have contributed through retaining dust-derived nutrients in the surface layer during the fall of 2013, promoting the development of several opportunistic phytoplankton groups in the western site M4.

7.  In contrast, persistently low coccolith fluxes in the central site M2, in particular of more opportunistic species, and the absence of major r-selected pulsed maxima, point to comparatively more stable and oligotrophic conditions prevailing in the centres of subtropical gyres.

Our findings (i) provide relevant evidence supporting the hypothesis of Saharan dust acting as a fertilizer for marine phytoplankton in the Atlantic Ocean, and (ii) highlight the importance of LPZ coccolithophore species in terms of coccolith
export production in the tropical Atlantic, with possible implications for the global oceanic carbonate budget.

## Acknowledgments

Sediment trap samples were collected by the Royal Netherlands Institute for Sea Research (NIOZ) in the framework of the ongoing multidisciplinary projects "TRAFFIC", funded by the Netherlands Organization for Scientific Research (NWO, project no. 822.01.008), and "DUSTTRAFFIC", funded by the European Research Council (ERC, project no. 311152).Lab



preparation of the 1/5th split of the original sediment trap sample was conducted at the NIOZ, whereas the splitting/filtering and SEM taxonomical analysis were performed at the Geosciences Dep. of the University of Bremen, Germany. The first author benefited from a Marie Curie post-doctoral fellowship attributed by the BREMENTRAC COFUND fellowship program. Regarding the satellite data used in this study, the authors would like to acknowledge the MODIS Atmosphere
Science Team, the Aerosol Retrieval Group and the MODIS Adaptive Processing System (MODAPS); the NASA EOSDIS Physical Oceanography Distributed Active Archive Centre (PO.DAAC) (http://podaac.jpl.nasa.gov/SeaSurfaceSalinity/Aquarius); and the Ocean Biology Processing Group (OBPG) and the Atmosphere Archive & Distribution System (LAADS) at NASA's Goddard Space Flight Centre. The CCMP Version-2.0 vector wind analyses were produced by Remote Sensing Systems (www.remss.com) and the CMAP Precipitation data were
provided by the NOAA/OAR/ESRL PSD, Boulder, Colorado, USA (http://www.esrl.noaa.gov/psd/). The authors are thankful to Leandro Ponsoni, Lluisa Cros, Oliver Knebel and Friederike Ebersbach for their contributions during the discussion of the data.

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
