# Peer review of "Coccolithophore fluxes in the open tropical North Atlantic: influence of thermocline depth, Amazon water and Saharan dust"

_Biogeosciences, 2017_

## Referee Comment (RC1) · A.J. Poulton (Referee) · 11 Jul 2017

**GENERAL COMMENTS**

The manuscript by Guerreiro et al. presents time-series records and analysis of coccolith fluxes from two sites in the central (sub-)tropical Atlantic Ocean, showing strong linkages to local hydrographic and climatic forcing on flux variability. Specifically, the authors show enhanced fluxes during periods of Saharan dust deposition, increased wind stress, and retroflection of the Amazon outflow. The manuscript is well written and clearly presented, with the data and interpretation supporting the conclusions drawn. Though it would have been slightly more informative to examine coccolith-calcite fluxes,

it is appreciated that this is likely to have only a minimal impact on the overall conclusions or insights gained. The article is a useful addition to the growing literature on coccolithophore ecology in low-nutrient environments and confirms the growing consensus that such regions of the ocean are not quiescent but are characterised by significant temporal (and spatial) variability. Before publication there are a few minor corrections that need attention.

SPECIFIC COMMENTS

Pg 6, Ln 14-16: Please note that the variability in incidental PAR levels at the two sites (range from 65.59 to 65.62 E m-2 d-1) is fairly minimal ($\sim$0.03 E m-2 d-1) and this should be mentioned in the text (despite the clear seasonality shown in Fig. 2).

Table 4: Please clarify the 'numbers' referred to in the table legend, e.g. "in Factor 3, numbers 7 and 12 indicate ..".

Pg 16, Ln 6: Please clarify 'Own SEM observations . . .'.

Pg 18, Ln 27 (and elsewhere): Consider including an explanation of K-/r-selected ecological strategies somewhere in the text for clarity.

Pg 18, ln 22-23: Consider rephrasing "In contrast, during spring, not only the Amazon was not yet present in the study area, but the nearly inexistent precipitation possibly resulted in . . ." - numerous negatives statements becomes a bit confusing.

Pg 18, ln 30-30: Unnecessary repetition of 'recently' in same sentence and Trichodesmium should be italicised (Trichodesmium).

Pg 18, ln 34: Correct spelling of Falkowsky to Falkowski (et al., 1998).

Pg 18, ln 39: Please explain meaning of LNLC (low-nutrient-low-chlorophyll) areas.

References: Couple of minor errors noticed, should all be checked. Those noticed are: Jickells et al. 2005. Please correct title. Margalef 1978 – not cited?

---

## Referee Comment (RC2) · Anonymous Referee #2 · 17 Jul 2017

General comments

The study by Guerreiro et al. report time-series coccolith fluxes to the deep ocean at two sediment trap sites in the tropical Atlantic and provide new insights in the environmental factors controlling their spatial and temporal variability. The authors demonstrate that the seasonality of the coccolith fluxes was strongly influenced by the location of the ITCZ and stability of the water column. They find similar seasonal flux pattern at both sites, but significantly higher coccolith fluxes at the western M4 site compared to the M2 site in the central Atlantic. Guerreiro et al. suggest that the higher fluxes at M4 compared to M2 are caused by the Saharan dust input and the eastward dispersion of

the Amazon River plume, which retains nutrients within its less dense surface waters. The study also demonstrates striking differences in the coccolithophore community structure between the sites with higher abundance of opportunistic coccolithophore specie at M4 site, which experiences sporadic nutrient inputs.

The sediment trap observations presented in the study are robust and supported well by the relevant remote sensing data providing meteorological, hydrographic and biogeochemical contexts. The statistical analysis and references to literature aiding the interpretation of the results and supporting the conclusions are also strong.

I recommend the publication the article by Guerreiro et al. in Biogesciences. Relatively minor issues however need to be addressed prior to publication of this article.

Specific comments

- The Abstract is at times too detailed (see comments for Page 1) with a couple of sentences being too long and thus hard to follow.

- The oceanographic and meteorological conditions at the trap location during October 2012-November 2013 are results but presented in the Methods section. I suggest moving lines 5-25 to the Results section.

- Table 1 is unnecessary as most of its content has been given in the main text. It can be moved to Supplementary material along with Table 2.

- Authors could provide more information on the splitting of the sediment trap samples (e.g. type of splitter, splitting error).

- The authors use $1° \times 1°$ box over each trap location to retrieve satellite-based meteorological and oceanographic data to support coccolith fluxes at M4 and M2. The authors should elaborate on their choice: is it based on the consideration of the 'statistical funnel' or particle catchment area of a moored-sediment trap (see e.g. Baker et al. 1988, Siegel et al. 1997, Waniek et al. 2000)? For example, do the authors have complementary data on current speeds above the traps to derive back-trajectories of

sinking particles? This would give some idea of the distance of particle origin in support of the chosen 1°×1° area.

My other comments related (mostly of technical/language nature) are outlined below:

Page1

Line 18: remove 'in successive 16-day intervals'

Line 19: correct to 'showed'.

Lines 24-25: provide coccolith fluxes as average ± stdev.

Line 27-30: split into two sentences.

Line 26: correct to 'contributed to higher fluxes'.

Line 36: replace 'an open-ocean tropical setting' with 'the tropical open ocean'

Lines 35-38: split into two sentences.

Line 13: replace 'from' to 'in'.

Line 14: remove 'next to'.

Lines 14-15: replace 'diazotrophic fixation . . ... by cyanobacteria' to simply 'N2 fixation by marine diazotrophs'.

Line 18: replace 'despite low . . .. per unit area' to 'despite relatively low primary production rates'.

Line 27-29: this sentence needs rephrasing, and possibly split into two sentences.

Line 30-31: I do not see the reason for the contrast with opportunistic coccolithophores in this sentence. I suggest splitting the sentence into two for clarity.

Lines 37: 'whereas. . . coccolithophores studies using sediment traps' is confusing.

Consider replacing to 'Although a significant amount of sediment trap data on coccol-ithophores fluxes exists for the open ocean. . .'.

Lines 8: replace 'to 75 m depth' with 'in the top 75 m'.

Lines 9-15: please itemise the depth floral groups defined by Poulton et al. 2017, otherwise the sentence appears to long and hard to follow.

Line 15: More insight into what?

Lines 1-3: this sentence needs rephrasing with respect to 'the world's largest river. . .. for discharging large volumes'. I suggest 'the world's largest river with respect to fresh water discharge into the open ocean'.

Lines 22-24: This sentence does not read well. Please rephrase. I suggest 'Two sediment traps at sites M2 14° N, 37° W) and M4 (12° N, 49° W) collected sinking particles at 1200 m depth in synchronous intervals of 16 days from October 19, 2012 to November 7, 2013'.

Lines 24-26: this sentence is more suitable for the Supplementary Information.

Lines 19-20: Does it mean that the larger the ratio the deeper nutricline is? Please, be a bit more specific in here.

Please move Table 2 to Supplementary information

Lines 5-25: This is result, please move to the appropriate section.

Line 15: Do you need these many significant figures for the PAR values? Please be

consistent with the number of significant digits when reporting values in the main text.

Line 11: replace 'commonest' to the 'most abundant'

Line 14: replace 'factors' temporal variability...' to 'assess temporal variability of the factors'.

Lines 18-20: I suggest to use mean ± stdev when reporting fluxes at the study sites. We can see the range from Figure 3 and Table 3.

Figure 3 – dashed line for the Shannon-Weaver Diversity Index cannot be seen or appears as a solid line.

Line 14: remove 'fairly'.

Line 4: this sentence is confusing, please rephrase. Figure 5 will benefit from segregating the coccolithophore taxa into LPZ and UPZ species (in the legend).

Lines 11 and 34: add subsection numbers for clarity.

Line 3:replace 'further west' to M4.

Table 3: identify LPZ and UPZ species; also, in table caption, consider replacing 'maximum' with 'range' as given in the table.

Figure 8: be consistent how you report units in the legend. I suggest removing word

'units' for precipitation rates. Also, in figure axes, consider using Latitude (°N) and Longitude (°W).

Line 5: replace 'drastically' to 'considerably' or 'significantly'.

Line 18: replace 'by contrast' to 'in contrast'.

Line 14: elaborate on the meaning of Factor 1 – what variables does it represent?

Line 17-19: this sentence does not read well, please consider rephrasing.

Line 8: remove 'quite'.

Line 34: correct 'Falkowsky' to 'Falkowski'.

Line 41: replace '…seem to fit quite nicely' with 'this fits well'.

---

## Referee Comment (RC3) · Anonymous Referee #3 · 8 Aug 2017

This works provides new insights into the ecological influences on coccolithophores especially the importance of the low photic zone taxa in tropical environments. The research is of interest because there have been very few papers on the topic especially from tropical sediment traps. The paper uses some colloquial non-scientific language and rambles on a bit too much. When reading the title I thought that the work would mainly deal with the implications of Amazon River and Saharan dust input on the distribution of coccolithophores, but this topic only comes in the last few pages and there is precious little written about this in the introduction. I think either change the title to a more general one or really emphasize the Amazon and dust contribution from the start and reduce all the other material (which I would suggest because it would make for a

much better read). One way to start would be to deemphasize the statistical aspect. I say this because the primary coccolithophore species reside in the lower photic zone while the environmental parameters are taken from second party satellite information which does not say much about the water column below the surface. Much of the statistics information are mostly broad conjectures. If the authors want to include the Statistica 13 data they should really state why it is so relevant. There should be much more information on the seasonal change in the water column. I don't think there is much insight from the satellite data . Much more interesting is actually the Amazon and dust story as the title suggests. The authors should move section 5.2.2 to the beginning of the discussion. Another issue is that the authors start by pointing out the similarity between the M2 and M4 trap data. Section 5.1 is dedicated to this. Later they go on at length about the differences (p20 lines 5-7 and figure 10). I suggest that the paper be reorganized and shortened, emphasizing the influence of the Amazon and dust.

Here are some editing changes that I suggest:

Page 1 line 17: North Atlanitc did not collect –misplaced modifier please reword

P1 line 20: "were yearlong dominated" >dominated throughout the year

P1-33: "Still> Nevertheless

P1-36: "seemingly similar" > not very scientific

P1-36: If they "differ greatly" they can't be "seemingly similar"

P2-2: "low productivity" throughout the water column or just at the surface?

P2-4: in "these" areas what does "these" refer to?

P2-9: "huge" amount ..huge is a colloquial term and not scientific

P2-19: "vastness" vague how about some numbers here

P3-9: "partially" is an ambiguous word

P4–2: "Large volumes" give numbers

P4: "The upper water masses in the study area" I suggest a figure showing these water masses and the locations of the traps particularly because most of the important species are located in the LPZ.

P4–12: "Further down up" could be better stated

P4_22: "19 October, 19 to & November 2013" please correct

P5–6: was "carried out"

P5-17: 'and to the total samples" > not clear

P5-22: "What about the other coccolithophore "deep water species"? there are more than 2

P5-23 3.3 Oceanographic and meteorological data. In table II the URLs are not sufficient. They should go down to the actual page locations that were used.

P6: too many "albeits"..and also one on p 9 ..some of them not accurately used.

P6-12; P6-13 try to use another word other than distinctive..which is not very scientific

P8 Fig 3: I don't see dashed lines for the Shannon-Weaver Diversity Index

P8 -23 are you sure about R. sessilis? Would like to see an image.

P9 fig 5 ..the color scheme could use improvement.. Hard to read because of the color choice ∫

---

## Author Response (AR1)

**Reply to Referee Comment #1 (Alex Poulton)**

The authors are most grateful to Alex Poulton for his overall positive feedback on our manuscript, and for the helpful specific comments, which will be all taken into account for the new updated version of the paper.

Reply to specific comments of Alex Poulton:

**AP: Pg 6, Ln 14-16: Please note that the variability in incidental PAR levels at the two sites (range from 65.59 to 65.62 E m-2 d-1) is fairly minimal (~0.03 E m-2 d-1) and this should be mentioned in the text (despite the clear seasonality shown in Fig. 2).**

CG: This part will be updated to:
"Despite of the narrow range of PAR values observed at both stations, a clear seasonal trend is evidenced from slightly higher PAR during spring and summer (up to 65.615 Einstein m$^{-2}$ d$^{-1}$ in April 2013) and slightly lower PAR during fall and winter (down to 65.588 Einstein m$^{-2}$ d$^{-1}$ in December 2012) at both stations (Fig. 2cd) "

**AP: Table 4: Please clarify the 'numbers' referred to in the table legend, e.g. "in Factor 3, numbers 7 and 12 indicate ..".**

CG: For clarification, the sample-numbers of the three pulsed flux events will be indicated in the plots of *E. huxleyi*, *G. oceanica* and *G. muellerae* from Figure 6, and properly explained in the caption of the figure. Regarding the caption of Table 4, it will be updated to:
"In Factor 3, numbers 7 and 12 indicate the sample reference for the time-intervals where *G. muellerae* and *E. huxleyi* produced "pulse-like" maxima at station M4 (i.e. late January and middle April, 2013, respectively) (see Fig. 6i,c), whereas in Factor 4, number 24 indicates the sample reference for the time-intervals where the "pulse-like" maxima of sea surface Chl-a and *G. oceanica*, together with a sharp increase of *E. huxleyi* occurred at station M4 (i.e. late October/early November 2013) (see Fig. 6c,j)."

**AP: Pg 16, Ln 6: Please clarify 'Own SEM observations . . .**

CG: We meant that during the SEM counting of coccoliths throughout sample M4-24, we have found (as referred in the text) extremely high amounts of large fragments of diatoms. Below you may see one of the pictures I took from the sample to illustrate this aspect. This will be written in a clearer way in the updated version of the manuscript.

[Figure]

**Pg 18, ln 22-23: Consider rephrasing "In contrast, during spring, not only the Amazon was not yet present in the study area, but the nearly inexistent precipitation possibly resulted in . . ." - numerous negatives statements becomes a bit confusing.**

CG: This will be updated to:
"In contrast, the Amazon plume was not yet present in the study area during spring and the nearly inexistent precipitation possibly resulted in comparably lower nutrient-bioavailability by dry dust deposition (see Ridame et al., 2014)."

**AP: Pg 18, Ln 27 (and elsewhere): Consider including an explanation of K-/r-selected ecological strategies somewhere in the text for clarity.**

CG: We will follow the recommendation of the referee by including a short definition of these terms in the introduction, pp. 2 line 30:
"Coccolithophores are amongst the most important phytoplankton groups within open-ocean, stratified-oligotrophic waters (e.g. Winter et al., 1994), hence displaying features more typical of K-selected taxa (Margalef, 1978). These so called "K-selected species" are better adapted to compete successfully for limited nutrient availability in more stable environments, such as tropical regions and subtropical gyres, presenting low maximum growth rates and fairly constant populations that are in equilibrium with the environmental resources (see Margalef, 1978). Still, coccolithophores also include more opportunistic (r- selected) taxa that quickly respond to short-term changes associated with nutrient input (e.g. Guerreiro et al., 2013). These so-called "r-selected species" are recognized for their ability to reproduce quickly in unstable conditions, being often characterized for higher maximum growth rates and well adapted to survive and flourish within nutrient-rich and turbulent coastal environments.

**AP: Pg 18, ln 30-30: Unnecessary repetition of 'recently' in same sentence and Tri-chodesmium should be italicised (*Trichodesmium*).**

CG: This will be updated to:
"High fluxes of organic material recently observed in a sediment trap in the North Atlantic subtropical gyre (23ºN 41ºW) have been associated to enhanced phytoplankton productivity resulting from stimulated nitrogen fixation by *Trichodesmium* species following the deposition of dust-derived nutrients (Pabortsava et al., 2017)."

**AP: Pg 18, ln 34: Correct spelling of Falkowsky to Falkowski (et al., 1998).**

CG: This will be properly corrected.

**AP: Pg 18, ln 39: Please explain meaning of LNLC (low-nutrient-low-chlorophyll) areas.**

CG: This will be updated to:
"(…) on the other hand, report on the importance of strong and short-term ("pulse-like") dust deposition events on marine productivity in Low Nutrient Low Chlorophyll (LNLC) areas (…)"

**AP: References: Couple of minor errors noticed, should all be checked. Those noticed are: Jickells et al. 2005. Please correct title. Margalef 1978 – not cited?**

CG: All the references will be checked including the ones referred by Alex Poulton, and Margalef (1978) will be added to the list of references.

**Reply to Referee Comment #2 (Anonymous)**

The authors are most grateful to the Anonymous Referee #2 for his overall positive feedback on the manuscript, and for the helpful specific comments, the majority of which will be taken into account for the new updated version of the paper.

Reply to specific comments:

**RC#2: The Abstract is at times too detailed (see comments for Page 1) with a couple of sentences being too long and thus hard to follow.**

CG: Thank you for the comment. The abstract will be reduced into a more focused and straightforward text in the updated (revised) version of the manuscript.

**RC#2: The oceanographic and meteorological conditions at the trap location during October 2012-November 2013 are results but presented in the Methods section. I suggest moving lines 5-25 to the Results section.**

CG: As suggested by the reviewer, this section will be moved to the beginning of the RESULTS (section 4.1)

**RC#2: Table 1 is unnecessary as most of its content has been given in the main text. It can be moved to Supplementary material along with Table 2.**

CG: The authors agree that Table 2 should be moved to the Supplementary Material, but kindly disagree of the suggestion to do the same with Table 1, due to its important auxiliary function to rapidly localize the reader concerning the traps' background information. To avoid unnecessary repetition of information regarding the two traps, the respective reference in the text will be shorten.

**RC#2: Authors could provide more information on the splitting of the sediment trap samples (e.g. type of splitter, splitting error).**

CG: As recommended by the referee, a short paragraph will be added to the beginning of section 3.2:
"Sediment trap samples from stations M4 and M2 were initially wet-sieved over a 1 mm

mesh, wet-split into five aliquot subsamples using a rotary splitter (WSD-10, McLane Laboratories), washed to remove the HgCl2 and salts, and centrifuged. Average weight differences between replicate aliquots were within 2.4 % (SD = 2.2), with 87 % of all samples differing < 5 % between splits (detailed procedure in Korte et al., 2017). Micropaleontological analysis was carried out from one 1/5 split of each original sample, after being oxidized in a low temperature asher for approximately 4h to remove the organic matter and obtain a sample strictly composed of mineral particles (Fallet et al. 2009)."

**RC#2: The authors use 1◦×1◦ box over each trap location to retrieve satellite-based meteorological and oceanographic data to support coccolith fluxes at M4 and M2. The authors should elaborate on their choice: is it based on the consideration of the 'statistical funnel' or particle catchment area of a moored-sediment trap (see e.g. Baker et al. 1988, Siegel et al. 1997, Waniek et al. 2000)? For example, do the authors have complementary data on current speeds above the traps to derive back-trajectories of sinking particles? This would give some idea of the distance of particle origin in support of the chosen 1◦×1◦ area.**

CG: Yes, each mooring was equipped with two Technicap PPS 5/2 sediment traps provided with a tilt meter at 1200 m (upper trap) and 3500 m (lower trap), two SBE MicroCat CTDs for conductivity, temperature and depth measurements, two Aanderaa RCM-11 current meters, and four floatation bodies to keep the mooring upright. The uppermost float included a downward-looking acoustic doppler current profiler (ADCP, 75Hz) for measuring current profiles (velocity and direction) and particle backscatter intensities, and a XEOS iridium beacon and flasher on top. Current-meter measurements showed that the average velocities around each mooring were < 6 cm s−1 in the deep and bottom ocean (at around 3500 and 4600 m water depth, respectively) and < 10 cm s−1 at around 1200 m water depth as measured by the ADCPs. Exception goes for only a few days when current velocities exceeded the 12 cm s−1 for the upper sediment traps at 1200 m at M2, and ~19 cm s−1 at 3350 m at M4, in early August 2013. All other sensors showed that the sediment traps at sites M2 and M4 remained well within 5° from the vertical and at constant depths during the entire sampling period. For the detailed description of the moorings and respective oceanographic instruments, the referee is referred to Korte et al. (2007).

The 2ºx2º box, corresponding to ~200kmx200km (1º = ~ 110 km), was assumed to be representative of the catchment area of a trap deployed at 1200 m depth, taking into account the sinking speed for marine phytoplankton and algal aggregates (as indicated, for example, in Table 1 of Waniek et al., 2000) and based on temperature-pressure measurements indicating that M2 and M4 were equally and effectively vertical during the deployment period. As recommended by the referee, the criteria for the selected box will be explained with more detail in the new version of the manuscript, and referring to the suggested references. Furthermore, we will rectify in the text that the box used in the manuscript was actually of 2ºx2º and not 1ºx1º as mistakenly indicated in the submitted version of the manuscript."

**My other comments related (mostly of technical/language nature) are outlined below:**
**Page1**
**Line 18: remove 'in successive 16-day intervals'**

CG: Ok

**Line 19: correct to 'showed'.**

CG: Ok

**Lines 24-25: provide coccolith fluxes as average ± stdev.**

CG: Ok

**Line 27-30: split into two sentences.**

CG: Ok

**Line 26: correct to 'contributed to higher fluxes'.**

CG: Ok

**Line 36: replace 'an open-ocean tropical setting' with 'the tropical open ocean'**

CG: Ok

**Lines 35-38: split into two sentences.**

CG: Ok

**Page 2**
**Line 13: replace 'from' to 'in'.**

CG: Ok

**Line 14: remove 'next to'.**

CG: This will be replaced by "In addition to"

**Lines 14-15: replace 'diazotrophic fixation . . ... by cyanobacteria' to simply 'N2 fixation by marine diazotrophs'.**

CG: Ok

**Line 18: replace 'despite low . . .. per unit area' to 'despite relatively low primary production rates'.**

CG: Ok

**Line 27-29: this sentence needs rephrasing, and possibly split into two sentences.**

CG: this will be rephrased to:
"Coccolithophores, being at the same time photosynthetic and calcifying, are major contributors to the organic and inorganic oceanic carbon pumps (e.g. Rost and Riebesell, 2004). Due to their ability to cover their cells with tiny calcite plates (the coccoliths), coccolithophores can be studied in time-series samples collected by deep-ocean sediment traps (e.g. Broerse et al., 2000; Sprengel et al., 2002; Ziveri et al., 1995; Koebrich et al., 2015), thus providing insight into the seasonal to inter-annual dynamics of open-ocean phytoplankton."

**Line 30-31: I do not see the reason for the contrast with opportunistic coccolithophores in this sentence. I suggest splitting the sentence into two for clarity.**

CG: This will be rephrased to:
Coccolithophores are amongst the most important phytoplankton groups within open-ocean, stratified-oligotrophic waters (e.g. Winter et al., 1994), hence displaying features more typical of K-selected taxa (Margalef, 1978). These so called "K-selected species" are better adapted to compete successfully for limited nutrient availability in more stable environments, such as tropical regions and subtropical gyres, presenting low maximum growth rates and fairly constant populations that are in equilibrium with the environmental resources (see Margalef, 1978). Still, coccolithophores also include more opportunistic (r- selected) taxa that quickly respond to short-term changes associated with nutrient input (e.g. Guerreiro et al., 2013). These so-called "r-selected species" are recognized for their ability to reproduce quickly in unstable conditions, being often characterized for higher maximum growth rates and well adapted to survive and flourish within nutrient-rich and turbulent coastal environments.

**Lines 37: 'whereas... coccolithophores studies using sediment traps' is confusing. Consider replacing to 'Although a significant amount of sediment trap data on coccolithophores fluxes exists for the open ocean. . .'.**

CG: Ok

**Page 3**
**Lines 8: replace 'to 75 m depth' with 'in the top 75 m'.**

CG: Ok

**Lines 9-15: please itemise the depth floral groups defined by Poulton et al. 2017, otherwise the sentence appears to long and hard to follow.**

CG: Ok

**Line 15: More insight into what?**

CG: The sentence will be rephrased to:
"Therefore, more insight into the ecological preferences and environmental needs required by coccolithophores can be expected from a clear distinction between the distribution and abundance of UPZ and LPZ taxa."

**Page 4**
**Lines 1-3: this sentence needs rephrasing with respect to 'the world's largest river. . .. for discharging large volumes'. I suggest 'the world's largest river with respect to fresh water discharge into the open ocean'.**

CG: Ok

**Lines 22-24: This sentence does not read well. Please rephrase. I suggest 'Two sediment traps at sites (M2 14◦ N, 37◦ W) and M4 (12◦ N, 49◦ W) collected sinking particles at 1200 m depth in synchronous intervals of 16 days from October 19, 2012 to November 7, 2013'.**

CG: Ok, and thank you!

**Lines 24-26: this sentence is more suitable for the Supplementary Information.**

CG: We agree.

**Page 5**
**Lines 19-20: Does it mean that the larger the ratio the deeper nutricline is? Please, be a bit more specific in here.**

CG: No, it is precisely the opposite: the larger the ratio (i.e. higher abundance of UPZ taxa), the shallower is the nutricline. This part of the text will be rephrased into a clearer manner.

**Page 6**
**Please move Table 2 to Supplementary information**

CG: Ok

**Lines 5-25: This is result, please move to the appropriate section.**

CG: Ok

**Line 15: Do you need these many significant figures for the PAR values? Please be consistent with the number of significant digits when reporting values in the main text.**

CG: The reason why PAR is referred to in higher number of digits compared to the other parameters/proxies is due to the fact that the range of values for PAR is comparably lower, in spite of its clear seasonal variation at both stations.

**Page 7**
**Line 11: replace 'commonest' to the 'most abundant'**

CG: Ok

**Line 14: replace 'factors' temporal variability...' to 'assess temporal variability of the factors'.**

CG: Ok

**Lines 18-20: I suggest to use mean ± stdev when reporting fluxes at the study sites. We can see the range from Figure 3 and Table 3.**

CG: Ok

**Page 8**
**Figure 3 – dashed line for the Shannon-Weaver Diversity Index cannot be seen or appears as a solid line.**

CG: The reviewer is correct: the line should be (and will) referred as solid.

**Line 14: remove 'fairly'.**

CG: Ok

**Page 9**
**Line 4: this sentence is confusing, please rephrase.**

CG: This sentence will be rephrased to:
"Most of the taxa produced much higher coccolith fluxes at the western station M4".

**Figure 5 will benefit from segregating the coccolithophore taxa into LPZ and UPZ species (in the legend).**

CG: OK

**Lines 11 and 34: add subsection numbers for clarity.**

CG: Ok

**Page 10**
**Line 3: replace 'further west' to M4.**

CG: Ok

**Page 11**
**Table 3: identify LPZ and UPZ species; also, in table caption, consider replacing 'max-imum' with 'range' as given in the table.**

CG: Ok

**Page 13**
**Figure 8: be consistent how you report units in the legend. I suggest removing word 'units' for precipitation rates. Also, in figure axes, consider using Latitude (◦N) and Longitude (◦W).**

CG: Ok

**Line 5: replace 'drastically' to 'considerably' or 'significantly'.**

CG: Ok

**Line 18: replace 'by contrast' to 'in contrast'.**

CG: Ok

**Page 15**
**Line 14: elaborate on the meaning of Factor 1 – what variables does it represent?**

CG: Factor 1 is the most important result from factor analysis, since it explains the highest percentage of the variability within the taxa and environmental parameters (i.e. 30%, compared to only 16%, 10% and 9% for F2, F3 and F4, respectively). That F1 is clearly characterized by the opposition between the central station M2 from the western station M4, it highlights the much higher abundances of *G. flabellatus* further west as the most important feature/difference between the two sites. This result indicates the presence of spatial variability shown by the LPZ flora as being the statistically more relevant factor explaining our flux records.
For clarification, a brief explanation will be included in this part of the discussion.

**Line 17-19: this sentence does not read well, please consider rephrasing.**

CG: This sentence will be rephrased to:
"Highest coccolith fluxes recorded under these conditions suggest that higher stability of the photic layer has favored the development and/or the settling of coccolithophores during these periods."

**Page 18**
**Line 8: remove 'quite'.**

CG: Ok

**Line 34: correct 'Falkowsky' to 'Falkowski'.**

CG: Ok

**Line 41: replace '. . .seem to fit quite nicely' with 'this fits well'.**

CG: Ok

**Reply to Referee Comment #3 (Anonymous)**

The authors are most grateful to the Anonymous Referee #3 for his overall positive feedback on the manuscript, and for the helpful specific comments, several of which will be taken into account for the new updated version of the paper.

**RC#3: The paper uses some colloquial non-scientific language and rambles on a bit too much.**

CG: The new version of the manuscript will take this comment into account.

**RC#3: When reading the tittle, I thought that the work would mainly deal with the implications of Amazon River and Saharan dust input on the distribution of coccolithophores, but this topic only comes in the last few pages and there is precious little written about this in the introduction. I think either change the title to a more general one or really emphasize the Amazon and dust contribution from the start and reduce all the other material (which I would suggest because it would make for a much better read).**

CG: After a long process of several tries in organizing the presented data, it is our conviction that the present structure is the more efficient in "telling the story" of our study. But the referee is correct when arguing that the title does not reflect the interesting result we have shown, for example, for the LPZ flora. To address this important remark, the title will be updated to:
"Coccolithophore fluxes in the open tropical North Atlantic: influence of thermocline depth, Amazon water and Saharan dust"

**RC#3: One way to start would be to deemphasize the statistical aspect. I say this because the primary coccolithophore species reside in the lower photic zone while the environmental parameters are taken from second party satellite information which does not say much about the water column below the surface. Much of the statistics information are mostly broad conjectures. If the authors want to include the Statistica 13 data they should really state why it is so relevant.**

CG: The authors agree when the referee argues that coccolithophores in highly stratified ocean conditions preferably thrive in deeper levels of the euphotic zone. Nevertheless, we don't have access to time-series data regarding the hydrological features of the euphotic

zone in the study area for the sediment trapping period. Hence, it is our conviction that the presented satellite data provide invaluable insight into several environmental processes which were reflected in terms of coccolith fluxes and species composition, and provide a more reliable and integrated interpretation of the coccolith flux data. Despite of the limitations of using statistical analyses including environmental data that only represent the atmospheric conditions and the surface of the ocean (of which we are fully aware), we have obtained a fairly good correspondence between surface satellite data and the trap records. A remarkable example of this regards the drastic salinity decrease and associated increase of Chl-a at the surface, during times of enhanced fluxes of dust, organic matter, *G. oceanica and E. huxleyi* (not to mention total mass fluxes and biogenic silica, as presented by Korte et al., 2017). On the basis of this we agree with the referee that this fair correspondence between surface satellite data and the trap records, as well as the rather satisfying result of 63% of explained variance should be (and will be) more stressed in the discussion of the paper.

**RC#3: There should be much more information on the seasonal change in the water column.**

CG: We acknowledge the importance of having more information regarding the upper water column conditions for a more accurate disclosure of the seasonal dynamics of the euphotic zone and related hydrological parameters. However, such water column monitoring was not performed for the studied period and hence we don't have such possibility.

**RC#3: I don't think there is much insight from the satellite data. Much more interesting is actually the Amazon and dust story as the title suggests. The authors should move section 5.2.2 to the beginning of the discussion.**

CG: The authors kindly disagree with the referee, as mentioned above.

**RC#3: Another issue is that the authors start by pointing out the similarity between the M2 and M4 trap data. Section 5.1 is dedicated to this. Later they go on at length about the differences (p20 lines 5-7 and figure 10). I suggest that the paper be reorganized and shortened, emphasizing the influence of the Amazon and dust.**

CG: The authors would like to stress that one of the most interesting aspects of this study is precisely in the seasonal similarity vs. spatial dissimilarity between the two stations, in terms of meteorological and hydrographic forcing, which is reflected by the coccolith fluxes and cocccolith settling assemblages. On the basis of this, it is our conviction that this aspect is actually one of the main findings of our study, and hence, should be properly highlighted. Nevertheless, the authors acknowledge that some parts of the manuscript could benefit from some textual shortening whereas the influence of the Amazon and dust could be more highlighted in the introduction.

Reply to specific comments:
**Page 1 line 17: North Atlanitc did not collect –misplaced modifier please reword**

CG: OK

**P1 line 20: "were yearlong dominated" >dominated throughout the year**

CG: OK

**P1-33: "Still> Nevertheless**

CG: OK

**P1-36: "seemingly similar" > not very scientific**

CG: "seemingly" will be replaced by "apparently"

**P1-36: If they "differ greatly" they can't be "seemingly similar"**

CG: Answered in the line above

**P2-2: "low productivity" throughout the water column or just at the surface?**

CG: low productivity throughout the water column but particularly at the surface. This will be rephrased.

**P2-4: in "these" areas what does "these" refer to?**

CG: The authors are referring to the centers of subtropical gyres and tropical open-ocean referred in the previous sentence. This will be rephrased into a clearer way.

**P2-9: "huge" amount ..huge is a colloquial term and not scientific**

CG: "huge" will be replaced by "massive" and concrete values will be referred

**P2-19: "vastness" vague how about some numbers here**

CG: The authors are convinced that, for the purpose of this merely introductory sentence, it is unnecessary to provide numbers. "Vastness" will be replaced by "large surface area"

**P3-9: "partially" is an ambiguous word**

CG: The authors meant to say that the results from Poulton et al. (2017) agree with a large part of the results from the previous authors, but not all of them.

**P4–2: "Large volumes" give numbers**

CG: Ok

**P4: "The upper water masses in the study area" I suggest a figure showing these water masses and the locations of the traps particularly because most of the important species are located in the LPZ.**

CG: Ok

**P4–12: "Further down up" could be better stated**

CG: This will be rephrased to "Further down to"

**P4_22: "19 October, 19 to & November 2013" please correct**

CG: The referee is correct, thank you!

**P5–6: was "carried out"**

CG: This will be replaced by "undertaken"

**P5-17: 'and to the total samples" > not clear**

CG: the authors meant to say "original sediment trap sample". This will be modified in the ms revision.

**P5-22: "What about the other coccolithophore "deep water species"? there are more than 2**

CG: The authors decided to exclude *R. sessilis* and only include *F. profunda* and *G. flabellatus* in this formula on the basis of the high abundance and very similar seasonal pattern between the latter two species.

**P5-23 3.3 Oceanographic and meteorological data. In table II the URLs are not sufficient. They should go down to the actual page locations that were used.**

CG: Thank you for this comment. The table will be updated to the following:

| Parameter | Acronym | Units | Sensor | Product | Resolution | Source | Reference |
|---|---|---|---|---|---|---|---|
| Chlorophyll | Chl-*a* | mg m$^{-3}$ | MODIS-Aqua | L2-daily | 1 km | https://oceancolor.gsfc.nasa.gov/cgi/browse.pl?sen=am | OBPG 2014; Hu et al. 2012 |
| | | | | L3-monthly | 4km | https://oceancolor.gsfc.nasa.gov/cgi/l3 | |
| Sea Surface Temperature | SST | °C | MODIS-Aqua | L2-daily | 1 km | https://oceancolor.gsfc.nasa.gov/cgi/browse.pl?sen=am | OBPG 2014; Frouin et al. 2002 & 2012 |
| Photosynthetic Available Radiation | PAR | Einstein m$^{-2}$ d$^{-1}$ | MODIS-Aqua | L2-daily | 1 km | https://oceancolor.gsfc.nasa.gov/cgi/browse.pl?sen=am | OBPG 2014; Brown and Minnett (1999) |
| Aerosol Optical Depth | AOD | none | MODIS-Aqua | L2-daily | 10 km | https://ladsweb.modaps.eosdis.nasa.gov/search/order/1/MYD04_L2--6 | Levy, R., Hsu, C., et al., 2015. |
| Precipitation | Prec | mm day$^{-1}$ | multiple | L3-daily | 0.25° | https://disc.gsfc.nasa.gov/datasets/TRMM_3B42RT_Daily_7/summary | Goddard Earth Sciences Data and Information Services Center (2016); Huffman et al. 2007; Xie and Arkin, 1997 |
| Sea Surface Salinity and Wind Speed | SSS WSp | none m s$^{-1}$ | Aquarius | L3-daily | 1° | https://podaac.jpl.nasa.gov/dataset/AQUARIUS_L3_WIND_SPEED_SMI_DAILY_V4 & https://podaac.jpl.nasa.gov/dataset/AQUARIUS_L3_SSS_SMI_DAILY_V4 | NASA Aquarius project, 2015a & 2015b; |

**P6: too many "albeits"..and also one on p 9 ..some of them not accurately used.**

CG: The authors will take this comment into account for the new updated version of the manuscript.

**P6-12; P6-13 try to use another word other than distinctive..which is not very scientific**

CG: "Distinctive" will be replaced by "distinct"

**P8 Fig 3: I don't see dashed lines for the Shannon-Weaver Diversity Index**

CG: The referee is correct. "Dashed line" will be replaced by "solid line".

**P8 -23 are you sure about R. sessilis? Would like to see an image.**

CG: The referee is invited to look at these two examples of coccoliths of *R. sessilis* (in the figure on the right, please note the coccolith on the upper left side)

[Figure]

**P9 fig 5 ..the color scheme could use improvement.. Hard to read because of the color choice**

CG: Ok

**List of all relevant changes made in the manuscript:**

- The title has been changed to "Coccolithophore fluxes in the open tropical North Atlantic: influence of thermocline depth, Amazon water and Saharan dust";

- Several textual changes have been made in the abstract;

- Min. and max. coccolith fluxes have been replaced by mean +/- StDev in the abstract and in the results;

- Several textual changes and a brief definition of r-selected and K-selected taxa has been included in section 1 (introduction);

- Several textual changes have been introduced in section 2 (Oceanographic and meteorological settings);

- Figure 1B has been added to previous Figure 1, to illustrate the main water masses described in section 2, based on temperature data from CTD profiling along a transatlantic array at 12ºN, and on water depths reported by Emery and Meincke (1986), Reid (1994) and Stramma and Schott (1999);

- A brief description regarding the first steps of lab preparation of the sediment trap samples, and several textual changes have been included in section 3.2 (Laboratory and microscope analysis);

- A brief explanation for choosing a 2ºx2º box for the satellite data has been included in section 3.3;

- former Table 2 has been moved the Supplementary Material and all the following tables were referenced in order accordingly;

- The description of the oceanographic and meteorological conditions formerly presented in section 3.3 is now presented in section 4.1 (Oceanographic and meteorological conditions during the sampling period); all the following sections within the "Results" have been reordered accordingly;

- Several textual changes have been introduced throughout section 4 (Results);

- The reference to samples M4-U7, M4-U12, M4-U24 and M2-U9 have been added to Figures 6c, &I and 6j, and properly explained in the respective caption;

- The LPZ taxa have been distinguished from the UPZ, both in Table 2 (former Table 3) and in the legend of Figure 5;

- An explanation for the references U7, U12, U24 and U9 has been added to the caption of Table 3 (Former table 4);

- Several textual changes have been introduced throughout section 5 (Discussion);

- Figure 8 has been slightly changed and the reference to Latitude (ºN) and Longitude (ºW) has been added to the YY and XX axis, respectively;

- A more detailed explanation for the importance of Factor 1 has been included in section 5.2.1 (Higher productivity and/or transfer efficiency of the LPZ flora at the western site M4);

- A final comment on the good correspondence obtained between the trap records and the satellite time-series data has been included in the part of the discussion regarding the "Influence of wind-forced water mixing and dispersal of the Amazon River Plume"

- The meaning of LNLC has been added to the part of the discussion regarding the "Influence of Saharan dust deposition"

[revised manuscript text omitted]